# BAP1/ASXL complex modulation regulates epithelial-mesenchymal transition during trophoblast differentiation and invasion

Vicente Perez-Garcia[1,2,3,4]*, Georgia Lea[1], Pablo Lopez-Jimenez[5], Hanneke Okkenhaug[1], Graham J Burton[2], Ashley Moffett[2,4], Margherita Y Turco[2,4], Myriam Hemberger[1,2,6,7]*

[1]Epigenetics Programme, The Babraham Institute, Babraham Research Campus, Cambridge, United Kingdom; [2]Centre for Trophoblast Research, Department of Physiology, Development and Neurosicence, University of Cambridge, Cambridge, United Kingdom; [3]Centro de Investigación Príncipe Felipe, Eduardo Primo Yúfera, Valencia, Spain; [4]Department of Pathology, University of Cambridge, Cambridge, United Kingdom; [5]Biology Department, Universidad Autonoma de Madrid, Madrid, Spain; [6]Department of Biochemistry and Molecular Biology, Cumming School of Medicine, University of Calgary, Calgary, Canada; [7]Alberta Children's Hospital Research Institute, University of Calgary, Calgary, Canada

*For correspondence:
vp379@cam.ac.uk (VP-G);
myriam.hemberger@ucalgary.ca
(MH)

**Competing interests:** The authors declare that no competing interests exist.

**Abstract** Normal function of the placenta depends on the earliest developmental stages when trophoblast cells differentiate and invade into the endometrium to establish the definitive maternal-fetal interface. Previously, we identified the ubiquitously expressed tumour suppressor BRCA1-associated protein 1 (BAP1) as a central factor of a novel molecular node controlling early mouse placentation. However, functional insights into how BAP1 regulates trophoblast biology are still missing. Using CRISPR/Cas9 knockout and overexpression technology in mouse trophoblast stem cells, here we demonstrate that the downregulation of BAP1 protein is essential to trigger epithelial-mesenchymal transition (EMT) during trophoblast differentiation associated with a gain of invasiveness. Moreover, we show that the function of BAP1 in suppressing EMT progression is dependent on the binding of BAP1 to additional sex comb-like (ASXL1/2) proteins to form the polycomb repressive deubiquitinase (PR-DUB) complex. Finally, both endogenous expression patterns and BAP1 overexpression experiments in human trophoblast stem cells suggest that the molecular function of BAP1 in regulating trophoblast differentiation and EMT progression is conserved in mice and humans. Our results reveal that the physiological modulation of BAP1 determines the invasive properties of the trophoblast, delineating a new role of the BAP1 PR-DUB complex in regulating early placentation.

## Introduction

The placenta is a complex organ essential for nutrient and oxygen exchange between the mother and the developing fetus. Normal placental function in humans depends on the earliest stages of development, when trophoblast cells proliferate and differentiate to form the villous tree and invade into the maternal decidua. Trophoblast invasion allows attachment of the placenta to the uterus and also mediates transformation of maternal spiral arteries, thereby ensuring an unimpeded blood flow into the intervillous space. This process is fundamentally important to secure an adequate supply of resources to the fetus. Several pregnancy complications such as miscarriage, pre-eclampsia, placenta

accreta, and fetal growth restriction (FGR) are underpinned by a primary defect in trophoblast invasion (*Brosens et al., 2011*; *Kaufmann et al., 2003*). Despite extensive research, the precise molecular mechanisms that regulate adequate trophoblast differentiation and invasion remain poorly understood.

As part of the Deciphering the Mechanisms of Developmental Disorders (DMDD) programme, we found that placental malformations are highly prevalent in embryonic lethal mouse mutants (*Perez-Garcia et al., 2018*). This means that a significant number of genetic defects that lead to prenatal death may be due to abnormalities of placentation. In addition, we identified new molecular networks regulating early placentation. One of these molecular hubs is centred around the tumour suppressor BRCA1-associated protein 1 (BAP1), a deubiquitinase enzyme involved in the regulation of the cell cycle, cellular differentiation, cell death, gluconeogenesis, and DNA damage response (*Carbone et al., 2013*). At the molecular level, BAP1 regulates a variety of cellular processes through its participation in several multiprotein complexes. Amongst others, BAP1 has been reported to interact with the BRCA1-BARD1 (BRCA1-associated RING domain I) complex, with forkhead box proteins K1 and K2 (FOXK1/2), host cell factor-1 (HCF-1), yin yang 1 (YY1), *O*-linked *N*-acetylglucosamine transferase (OGT), lysine-specific demethylase 1B (KDM1B) and methyl-CpG binding domain protein 5 and 6 (MBD5 and MBD6) (*Jensen et al., 1998*; *Misaghi et al., 2009*; *Baymaz et al., 2014*; *Dey et al., 2012*; *Nishikawa et al., 2009*; *Yu et al., 2010*).

BAP1 also binds the epigenetic scaffolding proteins additional sex combs-like-1/2/3 (ASXL1/2/3) to form the polycomb repressive deubiquitinase (PR-DUB) complex that exerts an essential tumour suppressor activity by regulating ubiquitination levels of histone H2A (H2AK119Ub) (*Scheuermann et al., 2010*). ASXL proteins are obligatory partners of BAP1 and this interaction is required for BAP1 activity (*Campagne et al., 2019*). Mutations and deletions in PR-DUB core subunits, *BAP1* and *ASXL,* are frequently associated with various malignancies (*Carbone et al., 2013*; *Murali et al., 2013*; *Abdel-Wahab et al., 2012*; *Triviai et al., 2019*; *Micol et al., 2014*).

In terms of its role in development, we have previously reported that *Bap1* knockout (KO) mouse conceptuses are embryonic lethal around midgestation (E9.5) and exhibit severe placental defects that likely contribute to the intrauterine demise (*Perez-Garcia et al., 2018*). Specifically, the placentas of *Bap1*-mutant conceptuses show defects in differentiation of the chorionic ectoderm into syncytiotrophoblast, a process required for the development of the labyrinth, the area of nutrient exchange in the mouse placenta. Although conditional reconstitution of gene function in the placenta but not the embryo did not rescue the intrauterine lethality, it substantially improved syncytiotrophoblast formation. Moreover, *Bap1*-mutant placentas show a striking overabundance of trophoblast giant cells (TGCs), the invasive trophoblast cell population. A similar bias towards the TGC differentiation pathway at the expense of the syncytiotrophoblast lineage was observed in *Bap1*-null mouse trophoblast stem cells (mTSCs), suggesting a critical role for BAP1 in regulating trophoblast biology (*Perez-Garcia et al., 2018*).

Recent reports have highlighted the possibility of co-evolution of shared pathways of invasion between trophoblast and cancer cells, in particular with regard to cell invasiveness and the capacity to breach basement membranes (*Kshitiz et al., 2019*; *Costanzo et al., 2018*). In both cases, the initial process is characterized by EMT where epithelial cells lose their polarity and cell-cell adhesion properties, and gain migratory and invasive properties of mesenchymal cells (*Parast et al., 2001*; *El-Hashash et al., 2010*). Understanding the mechanisms by which BAP1 regulates trophoblast differentiation and invasion will be important not only to uncover new molecular pathways involved in placental development, but also to shed light into the signalling pathways altered in tumours where BAP1 is mutated.

Here, we sought to determine the molecular mechanism by which BAP1 regulates trophoblast proliferation, differentiation, and invasion. Using CRISPR/Cas9-generated *Bap1⁻/⁻* mTSCs, we find that the deletion of *Bap1* does not affect their self-renewal capacity but precociously promotes the EMT process. Furthermore, we demonstrate that BAP1 downregulation is required to trigger EMT; consequently, *Bap1* overexpression, mediated by CRISPR-Synergistic Activation Mediator (SAM)-induced activation of the endogenous locus, slows down cell proliferation, delays TGC differentiation, and reduces trophoblast invasion. Analysis of the PR-DUB complex components *Asxl1* and *Asxl2* revealed that ASXL1 is downregulated in parallel to BAP1, regulating BAP1's stability. In contrast, *Asxl2* exhibits the opposite expression pattern, with a concomitant increase as cells differentiate. By knocking down *Asxl1* or *Asxl2* in mTSCs, we further show that BAP1 stability depends on its

interaction with ASXL proteins. Like $Bap1^{-/-}$ mTSCs, both *Asxl1*- and *Asxl2*-mutant mTSCs fail to induce syncytiotrophoblast differentiation. The functional characterization of BAP1 in the human placenta and human trophoblast stem cells (hTSCs) suggests that the role of BAP1 in regulating trophoblast differentiation and EMT progression is conserved in mice and humans. Indeed, overexpression of BAP1 in hTSCs indicates that BAP1 levels define the epithelial characteristics of hTSCs. Collectively, these data reveal a pivotal role of BAP1/ASXL complexes in regulating EMT as a requisite for trophoblast invasion and in regulating the finely tuned balance of lineage-specific differentiation into the various trophoblast subtypes.

## Results

### BAP1 is highly expressed in undifferentiated trophoblast and downregulated as cells enter the TGC lineage

To gain insight into the role of *Bap1* in trophoblast development, we first examined BAP1 expression in mTSCs. This unique stem cell type is derived from the trophectoderm of the blastocyst or from extraembryonic ectoderm (ExE) of early post-implantation conceptuses. mTSCs retain the capacity to self-renew and to differentiate into all trophoblast subtypes under appropriate culture conditions (*Tanaka et al., 1998*). Immunofluorescence analysis of BAP1 in mTSCs showed strong nuclear staining (*Figure 1A, B and C*). We noticed that mTSC colonies containing areas of spontaneous differentiation, identified by decreased ESRRB stem cell marker expression, displayed a concomitant reduction in BAP1 staining intensity, suggesting that BAP1 is downregulated as soon as trophoblast cells start to differentiate (*Figure 1B*). In line with these observations, differentiation of mTSCs in vitro revealed a significant reduction in BAP1 protein levels at days 3 and 6 of differentiation compared to stem cell conditions, as shown by immunofluorescence staining and Western blot (WB) analysis (*Figure 1C and D*). The strongest downregulation was seen at 6 days when giant cells are the prevailing differentiated cell type (*Murray et al., 2016*; *Perez-Garcia et al., 2018*). However, *Bap1* mRNA levels did not significantly change across this differentiation time course, indicating that the functional regulation of BAP1 takes place at the post-transcriptional level (*Figure 1E*).

To further corroborate these results, we performed BAP1 immunostainings on mouse conceptuses at day (E) 6.5 of gestation, a time window when the ExE is actively proliferating and differentiating into the ectoplacental cone (EPC). While ExE will go on to develop predominantly into the labyrinth at later stages of development, EPC cells will give rise to the placental hormone-producing spongiotrophoblast layer and to invasive TGCs (*Simmons et al., 2007*; *Woods et al., 2018*). Immunofluorescence analysis revealed strong nuclear BAP1 staining in the embryo proper (epiblast, Epi) and in the ExE. However, differentiating EPC showed a significantly reduced staining intensity (*Figure 1—figure supplement 1A, B and C*). In E9.5 placentae, BAP1 immunoreactivity was prominent in the developing labyrinth and spongiotrophoblast layer as well as in maternal decidual cells, but was markedly less pronounced in TGCs, again suggesting that BAP1 is specifically downregulated as trophoblast cells differentiate into TGCs (*Figure 1—figure supplement 1D*).

Overall, these results indicate that BAP1 is highly expressed in undifferentiated trophoblast of the ExE in vivo and in mTSCs in vitro. BAP1 is downregulated at the protein level specifically as cells enter the TGC lineage, suggesting a potential function of BAP1 in regulating trophoblast differentiation and invasiveness, a key property of TGCs.

### BAP1 deletion does not impair the stem cell gene-regulatory network

The proliferative and self-renewal capacity of mTSCs depends on FGF and Tgfβ1/activin A signalling pathways (*Tanaka et al., 1998*; *Erlebacher et al., 2004*). To further explore the main growth factor signals involved in regulating BAP1 protein levels, mTSCs were subjected to 3 days of differentiation in the presence of either FGF or conditioned medium (CM), which provides the main source of Tgfβ1/activin A in the complete TSC media. WB analysis showed that after 3 days of differentiation under standard differentiation conditions (base medium) or in the presence of CM, BAP1 was markedly downregulated. However, the presence of FGF alone maintained high BAP1 protein levels, indicating that FGF signalling is the main pathway driving BAP1 expression in stem cell conditions (*Figure 2A*).

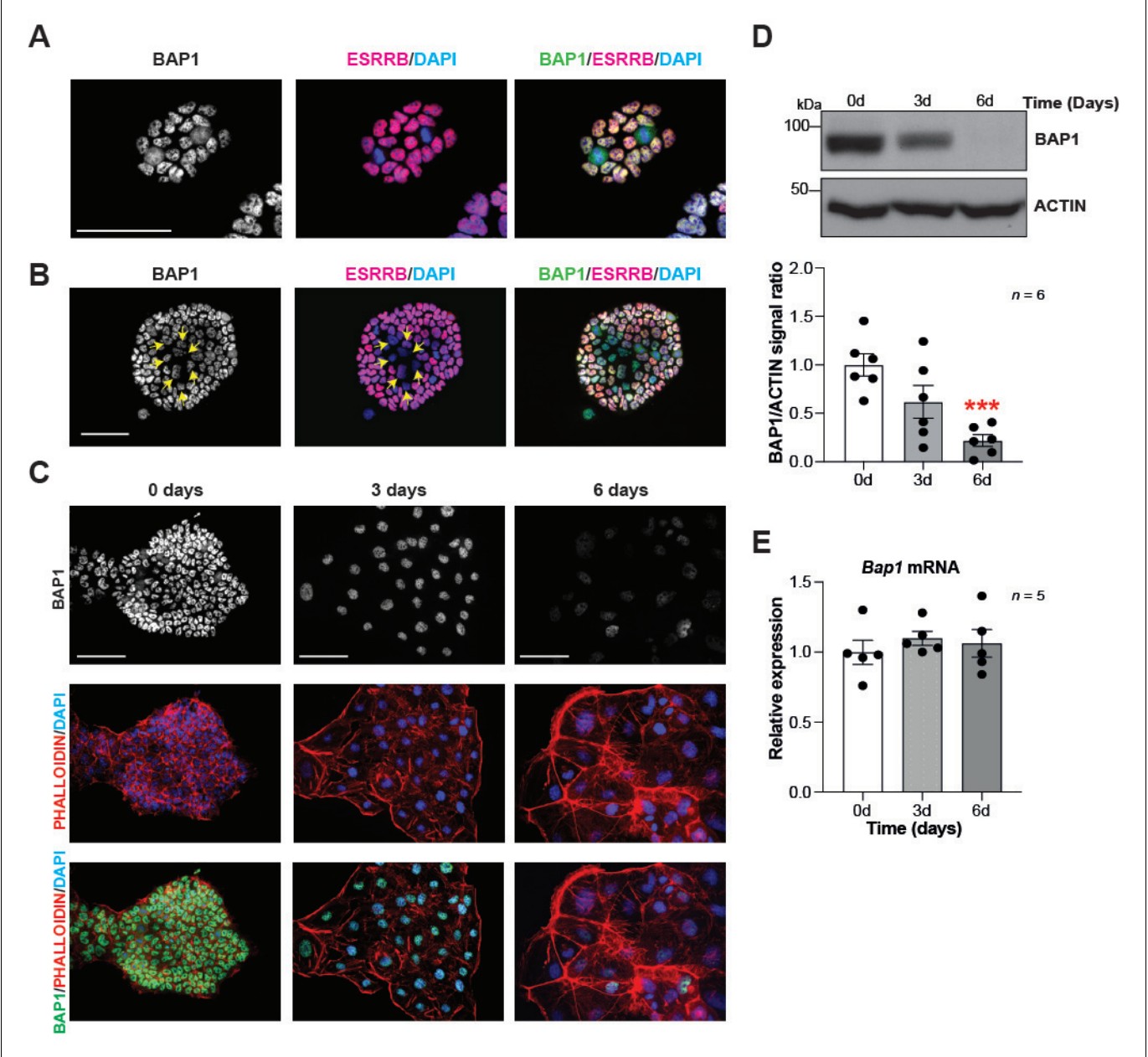

**Figure 1.** BAP1 protein levels are modulated during trophoblast differentiation. (**A, B**) Immunofluorescence staining of mouse trophoblast stem cells (mTSCs) in the stem cell state for BAP1 and the stem cell marker ESRRB. The strong nuclear BAP1 staining observed in mTSCs is slightly reduced in partially differentiated, ESRRB-low cells (arrows). Representative images of four replicates. Scale bar: 100 μm. (**C**) Immunofluorescence staining for BAP1 and F-actin with phalloidin of mTSCs, and of mTSCs differentiated for 3 and 6 days. BAP1 is downregulated as cells reorganize their cytoskeleton during trophoblast differentiation. Representative images of three replicates. Scale bar: 100 μm. (**D**) Western blot for BAP1 on mTSCs in the stem cell state and upon 3 days (3d) and 6 days (6d) of differentiation, confirming the downregulation of BAP1. Quantification of band intensities of six independent experiments is shown in the graph below. Data are normalized against ACTIN and represented relative to stem cell conditions (0d); mean ± SEM; ***p<0.001 (one-way ANOVA with Dunnett's multiple comparisons test). (**E**) RT-qPCR analysis of *Bap1* expression during a 6-day time course of mTSC differentiation shows that *Bap1* mRNA levels remain stable throughout the differentiation process. Expression is normalized to *Sdha* and displayed relative to stem cell conditions (0d). Data are mean of five replicates ± SEM (one-way ANOVA with Dunnett's multiple comparisons test). The online version of this article includes the following figure supplement(s) for figure 1:

**Figure supplement 1.** *Bap1* expression in early mouse placentation.

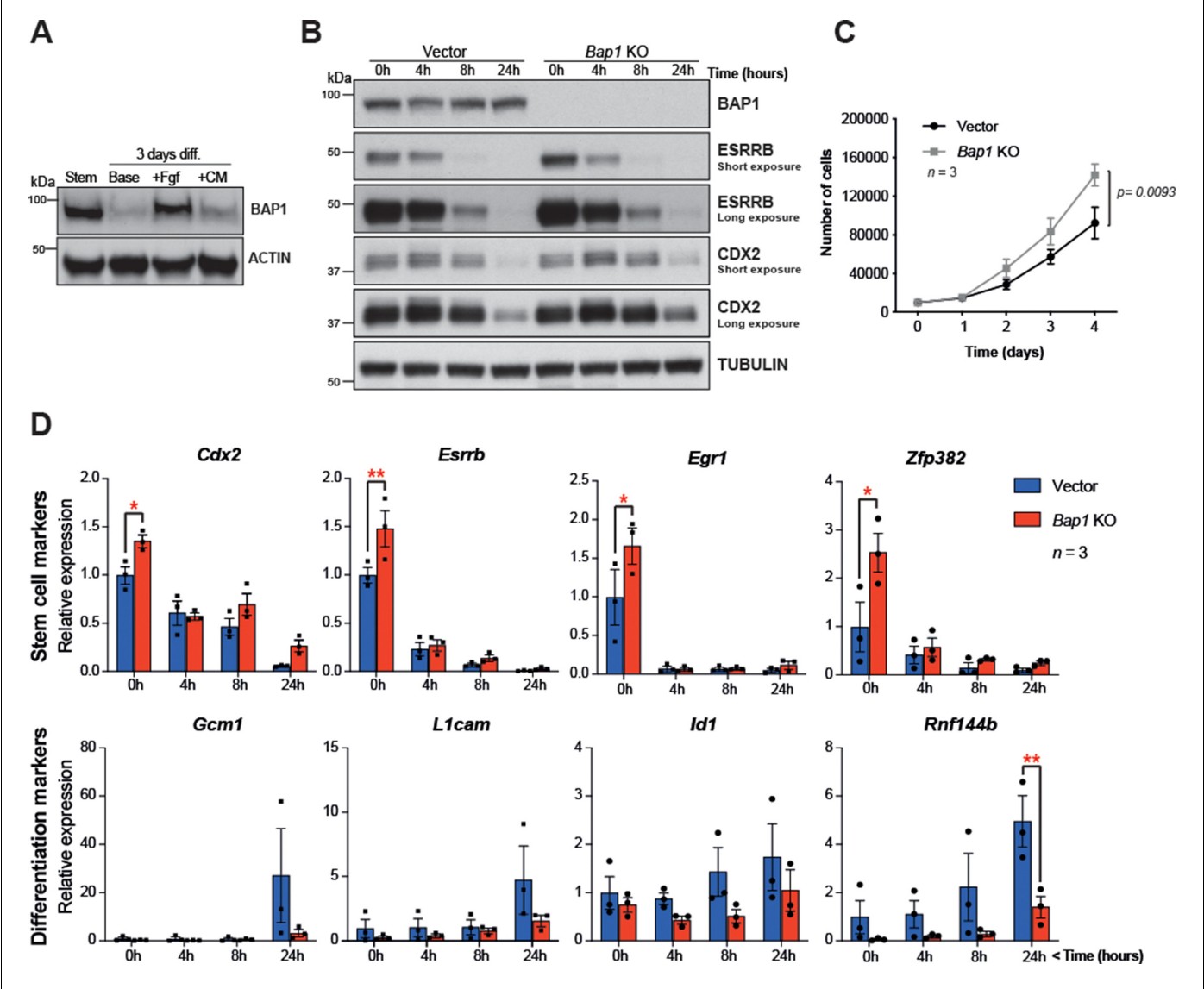

**Figure 2.** *Bap1* ablation does not negatively affect stemness. (**A**) Western blot analysis of mouse trophoblast stem cells (mTSCs) grown in stem cell conditions (Stem) and upon 3 day differentiation in standard base medium (Base), or in base medium supplemented with FGF or conditioned medium (CM). (**B**) Western blot analysis assessing the dynamic changes in the stem cell markers CDX2 and ESRRB across a short-term differentiation time course in vector control compared to *Bap1*-mutant mTSCs (stem cell conditions = 0 h, and differentiation at 4, 8, and 24 hours (h)). Blots are representative of two independent replicates. (**C**) Proliferation assay of control and *Bap1⁻/⁻* mTSCs over 4 consecutive days. *Bap1⁻/⁻* mTSCs exhibit a significant increase in the proliferation rate compared to vector control cells (mean ± SEM; n = 3). p=0.0093; two-way ANOVA with Holm-Sidak's multiple comparisons test. (**D**) RT-qPCR analysis of control and *Bap1⁻/⁻* mTSCs for stem cell and early differentiation marker genes. Stem cell markers are increased and the upregulation of differentiation markers delayed in *Bap1*-mutant mTSCs. Data are normalized to *Sdha* and displayed as mean of three biological replicates (i.e. independent clones) ± SEM; *p<0.05, **p<0.01 (two-way ANOVA with Sidak's multiple comparisons test).

The online version of this article includes the following figure supplement(s) for figure 2:

**Figure supplement 1.** *Bap1* ablation does not negatively affect stemness.

This raises the question whether the absence of BAP1 affects stem cell fate. The transcription factors CDX2 and ESRRB represent primary targets and direct mediators of FGF signalling in mTSCs (*Latos et al., 2015*) that are essential to keep mTSCs in a highly proliferative, undifferentiated state. Both factors are rapidly downregulated upon trophoblast differentiation (*Latos et al., 2015*; *Luo et al., 1997*; *Strumpf et al., 2005*). Previously, we reported that *Bap1* deletion in mTSCs resulted in an upregulation of *Cdx2* and *Esrrb* mRNA levels (*Perez-Garcia et al., 2018*). To further

investigate the effect of BAP1 on stem cells markers, we assessed CDX2 and ESRRB protein levels in *Bap1^{-/-}* mTSCs compared to (empty vector) control cells across 24 hr of differentiation (0h = stem cell conditions; 4h, 8h, 24h = hours upon differentiation). The absence of BAP1 resulted in increased ESRRB and CDX2 protein levels, thus confirming our previous observations. *Bap1* deficiency also increased proliferation rates in stem cell conditions (*Figure 2B and C*). The higher residual expression of ESRRB and CDX2 proteins detected after 24h of differentiation may indicate a potential delay in mTSC differentiation (*Figure 2B*). Indeed, analysis of the mRNA expression dynamics of the trophoblast stem cell markers *Cdx2, Esrrb, Egr1,* and *Zpf382* indicated that *Bap1*-mutant mTSCs differentiated more slowly than control counterparts during the initial 24h of differentiation (*Figure 2D* and *Figure 2—figure supplement 1A*). In line with these results, we also observed that the upregulation of early mTSC differentiation markers such as *Gcm1, L1cam, Id1,* and *Rnf44b* was delayed in *Bap1^{-/-}* mTSCs compared to control cells (*Figure 2D* and *Figure 2—figure supplement 1A*).

## *Bap1^{-/-}* mTSCs undergo EMT

The appearance of *Bap1^{-/-}* TSCs under phase contrast revealed a phase bright, refractile, and loosely associated morphology with poor cell-cell contacts. This was in contrast to the colonies of vector control cells, suggesting that they may have undergone an EMT-like transition (*Figure 3A*), known to occur when trophoblast differentiates towards the invasive TGC lineage (*Sutherland, 2003*). The morphology of *Bap1^{-/-}* mTSC colonies led us to hypothesize that BAP1 affects EMT in trophoblast. To investigate this, we studied the global expression profile of *Bap1*-mutant mTSCs compared to control cells in stem cell conditions (0d) and upon 3 days of differentiation (3d). Unbiased clustering and principal component analysis (PCA) clearly showed that the differentially expressed genes (DEG) were determined by the absence of BAP1 and by the day of differentiation (*Figure 3B* and *Figure 3—figure supplement 1A*). Gene ontology analysis revealed an enrichment of genes involved in regulation of extracellular matrix, cell junction, and cell adhesion at both time points analysed, concordant with the marked change in morphology of *Bap1^{-/-}* mTSCs (*Figure 3C*, *Figure 3—figure supplement 1B* and *Supplementary file 1* and *2*). In line with these results, *Bap1^{-/-}* mTSCs exhibited a significant decrease in cell adhesion on tissue culture plastic (*Figure 3D*), which was even more obvious when *Bap1^{-/-}* mTSCs were grown in 3D organoid-like trophospheres (*Figure 3E*; *Rai and Cross, 2015*).

At the molecular level, the reduction in cell adhesion correlated with a significant downregulation of E-cadherin (*Cdh1*), an epithelial hallmark, in *Bap1^{-/-}* compared to vector control cells (*Figure 3F*, *Figure 3—figure supplement 1D* and *Supplementary file 1* and *2*). Stringent calling (DESeq2 and intensity difference analysis) of DEG revealed that several genes involved in the stabilization of cell-cell contacts and epithelial integrity (*Claudin 4* [*Cldn4*], *Claudin 7* [*Cldn7*], *Desmoplakin* [*Dsp*], and *Serpine1*) were downregulated in *Bap1*-mutant mTSCs (*Figure 3—figure supplement 1C* and *Supplementary file 1*). Concomitant with the downregulation of epithelial markers like *Cdh1*, mesenchymal markers including N-cadherin (*Cdh2*), *Zeb2*, and *Vimentin (Vim)* were upregulated in the absence of *Bap1* (*Figure 3F*). These data indicate that *Bap1*-null mTSCs display a pronounced and precocious EMT phenotype.

TGC formation is characterized by cytoskeletal rearrangements, exit from the cell cycle, DNA endoreduplication, and production of trophoblast-specific proteins such as placental prolactins. Thus, undifferentiated trophoblast cells exhibit little organized actin and few peripheral focal complexes, whereas TGCs show a highly organized cytoskeleton containing prominent actin stress fibres linked to gain in motility and invasiveness (*Parast et al., 2001*; *El-Hashash et al., 2010*). As expected from the mRNA expression analysis, *Bap1^{-/-}* mTSCs showed a loss of membrane-associated CDH1 staining and disorganized cytoskeleton in stem cell conditions, with increased numbers of actin stress fibres upon differentiation, suggesting a more TGC-like and invasive phenotype compared to wild-type (vector) cells (*Figure 3G*). Indeed, *Bap1^{-/-}* TSCs were also more invasive through extracellular basement membrane (Matrigel) compared to vector control mTSCs in Transwell invasion experiments (*Figure 3H and I*). In line with these results, the DEG in *Bap1^{-/-}* mTSCs showed significant overlap with the gene expression signatures of tissues prone to form tumours such as *Bap1^{-/-}* melanocytes and mesothelial cells (*He et al., 2019*; *Figure 3—figure supplement 1E*). Altogether, these results indicate that the lack of *Bap1* triggers EMT in mTSCs that recapitulates critical aspects of early malignant transformation.

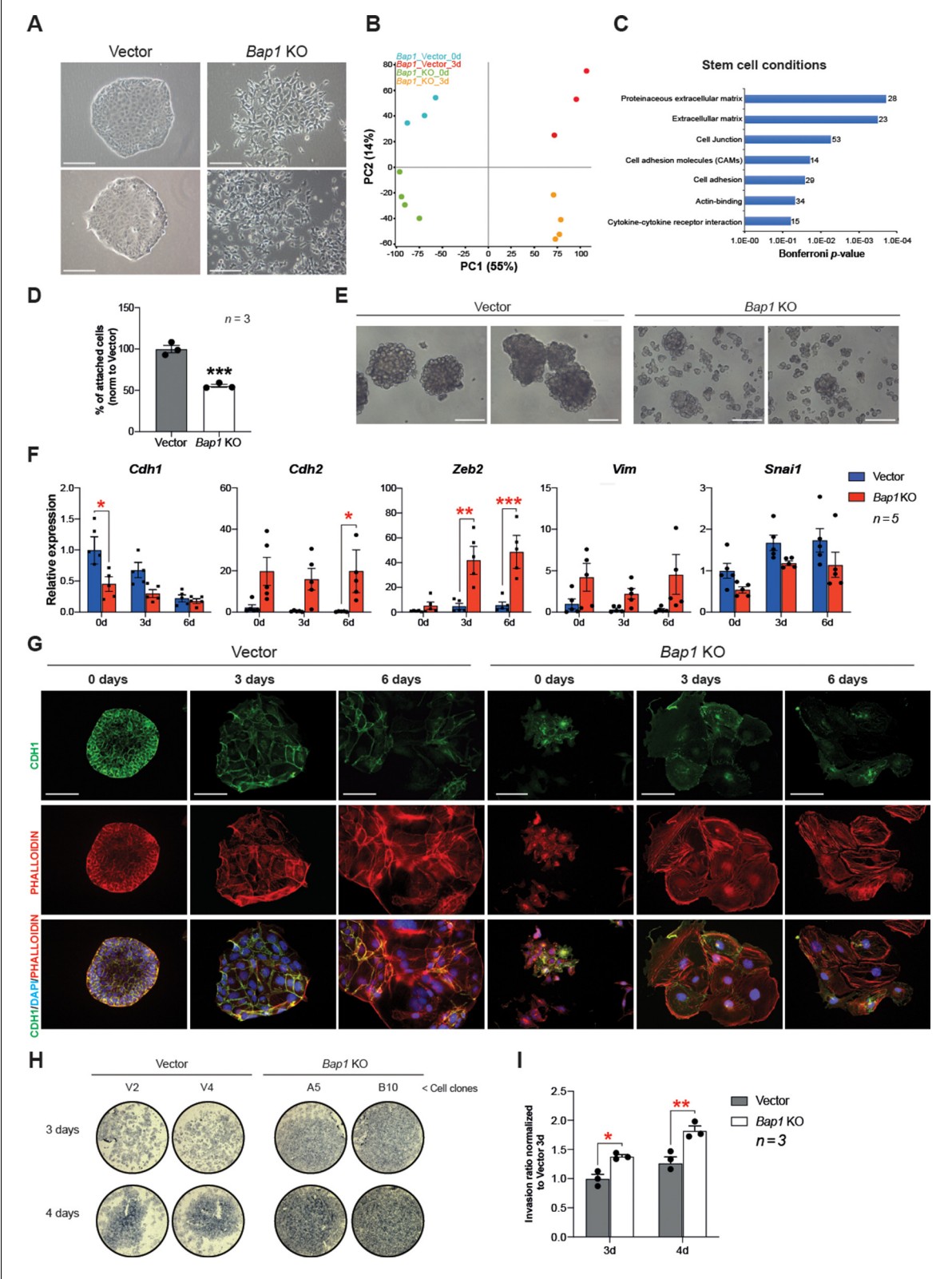

**Figure 3.** *Bap1* deficiency promotes epithelial-mesenchymal transition (EMT). (**A**) Colony morphology of wild-type (vector) and *Bap1*-mutant mouse trophoblast stem cells (mTSCs). *Bap1*$^{-/-}$ mTSCs show a fibroblast-like morphology with loss of cell-cell attachment compared to vector control mTSCs. Images are representative of five independent TSC clones each. Scale bar: 100 μm. (**B**) Principal component analysis of global transcriptomes of independent vector control (n = 3) and *Bap1* knockout (KO) (n = 4) clones grown in stem cell conditions (0d) and after 3 days of differentiation (3d). (**C**) *Figure 3 continued on next page*

*Figure 3 continued*

Gene ontology analyses of genes differentially expressed between vector and *Bap1*-mutant mTSCs in stem cell conditions. (D) Cell adhesion assay showing that *Bap1*-mutant mTSCs are less well attached to cell culture plastic compared to vector control cells. Data are mean of three independent replicates with three biological replicates ( = independent clones) per experiment. ***p<0.001 (Student's t-test). (E) Morphology of 3D-trophospheres after 8 days of differentiation in low attachment conditions. Representative images of 2 independent vector control and *Bap1* KO cell clones. Scale bar: 200 μm. (F) RT-qPCR analysis of EMT marker expression during a 6-day differentiation time course. Data are normalized to *Sdha* and displayed relative to vector in stem cell conditions (0d). Data are mean of five biological replicates (i.e. independent clones) ± SEM; *p<0.05, **p<0.01, ***p<0.001 (two-way ANOVA with Sidak's multiple comparisons test). (G) Immunofluorescence analysis for CDH1 and F-actin (phalloidin) of vector control and *Bap1*-mutant mTSCs over 6 days of differentiation. Lack of BAP1 reduces cell-cell junctions (CDH1 staining) with a profound reorganization of the cytoskeleton (increased actin stress fibres). Data are representative of five independent vector control and *Bap1* KO clones each. Scale bar: 100 μm. (H) Transwell invasion assay of vector control (V2, V4) and *Bap1*-mutant (clones A5, B10) mTSCs after 3 and 4 days of differentiation. Photographs of invasion filters show haematoxylin-stained cells that reached the bottom side of the filter after removal of the reconstituted basement membrane matrix (Matrigel). (I) Quantification of invaded cells, measured by colour intensity, normalized to 3-day controls. Data are mean of three independent replicates (three biological clones in each replicate) ± SEM; *p<0.05, **p<0.01** (two-way ANOVA with Sidak's multiple comparisons test).

The online version of this article includes the following figure supplement(s) for figure 3:

**Figure supplement 1.** *Bap1* deficiency induces precocious epithelial-mesenchymal transition (EMT) with features of malignant transformation.

## BAP1 downregulation is critical to trigger EMT during trophoblast differentiation

In order to confirm that BAP1 is one of the main regulators of EMT during trophoblast differentiation, we overexpressed *Bap1* using the CRISPR/gRNA-directed Synergistic Activation Mediator (SAM) technology (*Konermann et al., 2015*). One out of three single guide RNAs (sgRNAs) tested induced robust upregulation of *Bap1* mRNA and BAP1 protein levels compared to mTSCs transduced with a non-targeting sgRNA (NT-sgRNA) (*Figure 4A* and *Figure 4—figure supplement 1A*). The upregulation of *Bap1* resulted in tight epithelial mTSC colonies that proliferated at a slower rate than NT-sgRNA control mTSCs (*Figure 4B and C*). To gain insight into the global transcriptional response to *Bap1* overexpression, we performed RNA-seq on mTSCs grown in stem cell conditions (0d) and after 3 days of differentiation (3d). PCA showed that, besides the growth conditions, samples clearly cluster by the levels of BAP1 within the cells (*Figure 4—figure supplement 1B and C*). A stringent assessment of the deregulated genes (DESeq2 and intensity difference filter) revealed that, in addition to *Bap1*, a cohort of 80 genes were significantly deregulated in stem cell conditions with a robust upregulation of genes involved in cell junction biology and maintenance of epithelial integrity – such as *Plakophilin 2* (*Pkp2*), *Keratin-7/8/19* (*Krt7/8/19*), *Desmoplakin* (*Dsp*), and *Cingulin* (*Cgn*) (*Figure 4D* and *Supplementary file 3* and *4*). In line with these observations, gene ontology analysis revealed an overrepresentation of extracellular matrix and cell adhesion molecules, suggesting an increase in epithelial features of BAP1-overexpressing cells compared to control cells (*Figure 4E* and *Supplementary file 3* and *4*). Intriguingly, there was substantial overlap between genes downregulated in *Bap1* KO mTSCs and those upregulated in *Bap1*-overexpressing cells, and conversely also between genes upregulated in the KO and downregulated in the overexpressing cells. Thus, the two opposing models of *Bap1* modulation (KO vs. overexpression) provided mirror-image results that pivoted around the biological processes of epithelial cell integrity, cell adhesion, and cytoskeletal organization (*Figure 4—figure supplement 1E* and *Supplementary file 5*).

To corroborate these RNA-seq results, we validated several EMT markers by RT-qPCR and confirmed that the overexpression of *Bap1* induced a significant upregulation of *E-cadherin* (*Cdh1*) with concomitant downregulation of *N-cadherin* (*Cdh2*) and *Vimentin* (*Vim*) (*Figure 4F*). In line with the re-acquisition of epithelial properties, *Bap1*-overexpressing mTSCs exhibited a delay in differentiation towards the invasive TGC lineage and lower invasive capacity through Matrigel compared to NT-sgRNA control cells (*Figure 4G and H* and *Figure 4—figure supplement 1F*). Finally, we corroborated the data obtained by CRISPR/Cas9-SAM overexpression by performing exogenous GFP-*Bap1* overexpression experiments in mTSCs grown in stem cell conditions, which similarly resulted in a significant upregulation of *Cdh1* and strong downregulation of *Cdh2*, *Zeb1*, *Zeb2*, *Snai1*, and *Vim* expression (*Figure 4—figure supplement 1G and H*). These results demonstrate that precise levels of BAP1 regulate mTSC morphology, and that modulation of BAP1 levels affects the extent and speed at which trophoblast cells undergo EMT. Altogether these results indicate that the downregulation of BAP1 is critical for triggering EMT and invasiveness of trophoblast cells.

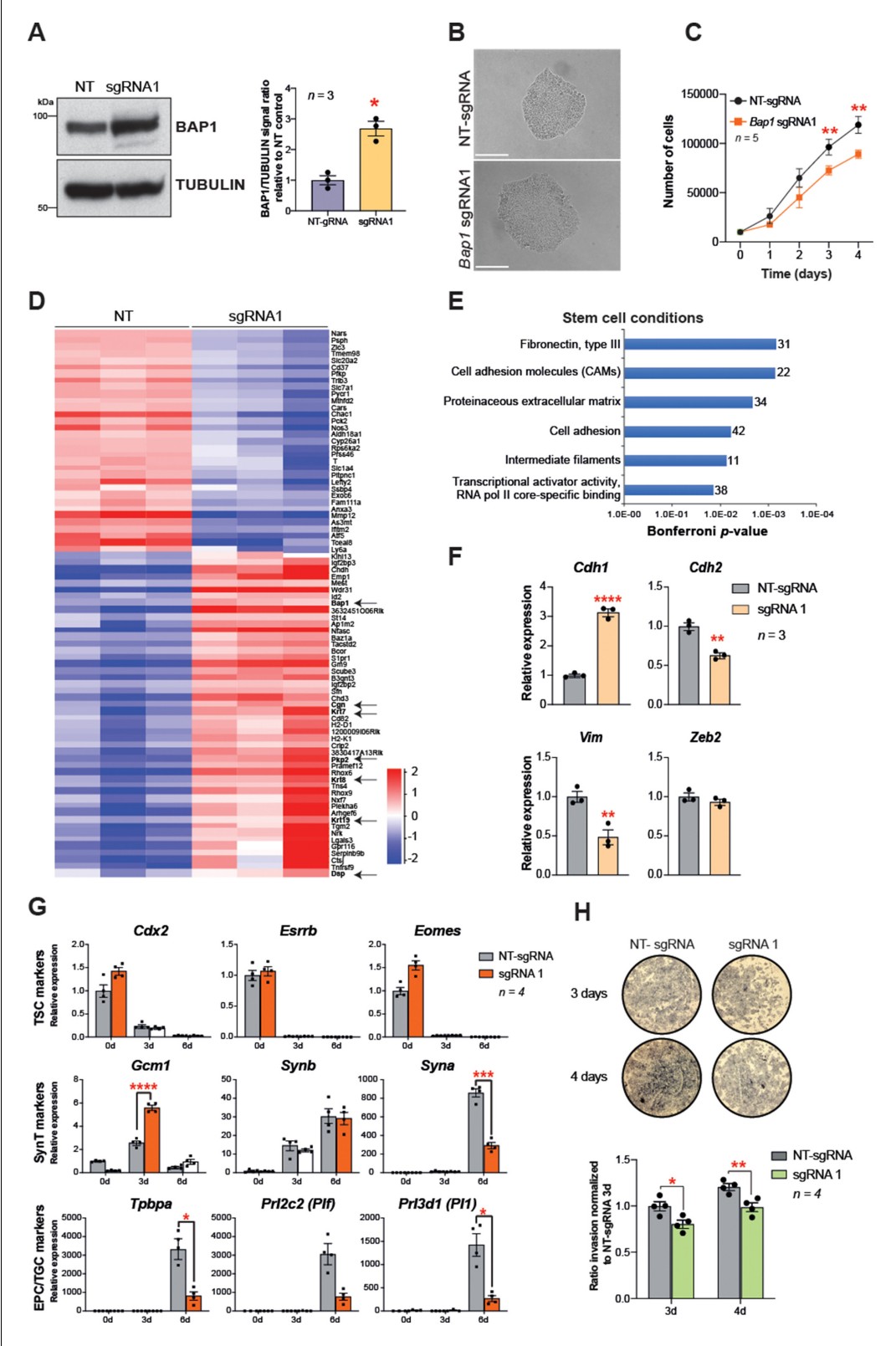

**Figure 4.** *Bap1* overexpression enhances epithelial features and reduces invasiveness. (**A**) Western blot analysis to confirm the overexpression of *Bap1* in mouse trophoblast stem cells (mTSCs) induced by transduction of the gene-activating single guide RNA one (sgRNA1) compared to non-targeting sgRNA (NT-sgRNA). TUBULIN was used as loading control. Graph shows the quantification of three independent replicates. Data are mean ± SEM; *p<0.05 (Student's t-test). (**B**) Colony morphology of NT-sgRNA and sgRNA1-transduced mTSCs. Overexpression of BAP1 in sgRNA1 mTSCs increases

*Figure 4 continued on next page*

*Figure 4 continued*

epithelioid features of the cell colonies. (C) Proliferation assay of control NT-sgRNA and sgRNA1 *Bap1*-overexpressing mTSCs over 4 consecutive days. sgRNA1 mTSCs exhibit a significant decrease in the proliferation rate compared to NT-sgRNA control cells (mean ± SEM; n = 5 each). **p<0.01; two-way ANOVA with Holm-Sidak's multiple comparisons test. (D) Heatmap of mean row-centred log$_2$ RPKM values of differentially expressed genes (DESeq2 and intensity difference) in mTSCs transduced with NT-sgRNA compared to sgRNA1. Arrows point to *Bap1* itself and to genes associated with the reinforcement of epithelial integrity. Three independent biological replicates per genotype were sequenced. (E) Gene ontology analysis of genes differentially expressed between sgRNA1 and NT-sgRNA mTSCs grown in stem cell conditions. (F) RT-qPCR analysis of epithelial and mesenchymal markers in NT-sgRNA control cells compared to sgRNA1 *Bap1*-overexpressing mTSCs. Data are normalized to *Sdha* and are displayed as mean of three replicates ± SEM; **p<0.01, ****p<0.0001 (Student's t-test). (G) Analysis of NT-sgRNA and sgRNA1 mTSCs grown in self-renewal conditions (0d) or after differentiation for 3 and 6 days (d) assessed by RT-qPCR. Data are mean ± SEM of n = 4 independent replicates. *p<0.05, **p<0.01, ***p<0.001, ****p<0.0001 (two-way ANOVA with Sidak's multiple comparisons test). (H) Transwell invasion assays of NT-sgRNA control and *Bap1*-overexpressing mTSCs. Representative images are shown. Quantification of invaded cells, measured by the colour intensity, normalized to 3-day NT-sgRNA. Data are mean of four independent replicates ± SEM; *p<0.05, **p<0.01 (two-way ANOVA with Sidak's multiple comparisons test).

The online version of this article includes the following figure supplement(s) for figure 4:

**Figure supplement 1.** *Bap1* overexpression increases epithelial features of mouse trophoblast stem cells (mTSCs).

## BAP1 and ASXL1/2 complexes are co-regulated during trophoblast differentiation

Interaction of BAP1 with ASXL proteins promotes its stability and enzymatic activity (*Campagne et al., 2019*). In order to investigate the role of the BAP1:ASXL complex in regulating trophoblast biology, we first examined gene expression of ASXL family members *Asxl1* and *Asxl2* over a 6-day differentiation time course. RT-qPCR and WB analysis showed that ASXL1 was highly expressed in mTSCs under stem cell conditions and strongly downregulated during trophoblast differentiation, in parallel to decreasing BAP1 protein levels. ASXL2 expression displayed the opposite trend with maximal levels in differentiated trophoblast (*Figure 5A and B*). This expression pattern was further validated by immunofluorescence (*Figure 5—figure supplement 1A and B*).

These results prompted us to investigate the nature of the BAP1-ASXL interaction in the mTSC context. To study the endogenous association of BAP1 and ASXL, we immunoprecipitated BAP1, ASXL1, and ASXL2 from extracts of mTSCs grown in stem cell conditions and tested for reciprocal interactions by WB. While we were not able to detect an association of BAP1 and ASXL2 in stem cell conditions, co-immunoprecipitation of BAP1 and ASXL1 revealed that BAP1:ASXL1 is the predominant complex in mTSCs (*Figure 5C* and *Figure 5—figure supplement 1C*). Then, we further analysed whether this interaction regulates the stability of the BAP1 and ASXL proteins. Whereas the absence of BAP1 did not affect the stability of ASXL1 and ASXL2 (*Figure 5—figure supplement 1D*), small interference RNA (siRNA)-mediated knockdown of either ASXL1 or ASXL2 resulted in a decrease of BAP1 protein levels (*Figure 5D*). This was particularly significant in the case of ASXL1 knockdown, in line with ASXL1 being the major complexing partner of BAP1 in stem cell conditions (*Figure 5D*).

To gain further insight into the specific roles of BAP1's interaction partners, we generated *Asxl1* and *Asxl2* KO mTSCs using CRISPR/Cas9 technology (*Figure 5—figure supplement 1E*). The deletion of *Asxl1* or *Asxl2* did not affect stemness. However, under differentiation conditions, *Asxl1*$^{-/-}$ and *Asxl2*$^{-/-}$ mTSCs failed to upregulate markers of syncytiotrophoblast, whereas the differentiation towards TGCs was promoted (*Figure 5F and G* and *Figure 5—figure supplement 1F*). This defect phenocopied the syncytiotrophoblast differentiation defect we had previously reported for *Bap1*-mutant cells (*Perez-Garcia et al., 2018*). Moreover, the absence of *Asxl1* and *Asxl2* induced an upregulation of EMT markers such *Cdh2, Vim, Zeb1,* and *Zeb2*, suggesting that ASXL1 and ASXL2 together with BAP1 contribute to the modulation of EMT as a critical process during trophoblast differentiation (*Figure 5—figure supplement 1F*).

## BAP1 PR-DUB complex is also regulated during human trophoblast differentiation

TGCs represent the invasive trophoblast cell type in mice whereas in humans, this function is exerted by extravillous trophoblast (EVT). As in mouse, the gain of invasive properties is accompanied by an EMT process (*DaSilva-Arnold et al., 2015*; *E Davies et al., 2016*; *Vićovac and Aplin, 1996*). Polycomb group complexes, including the BAP1 PR-DUB, are well conserved throughout evolution

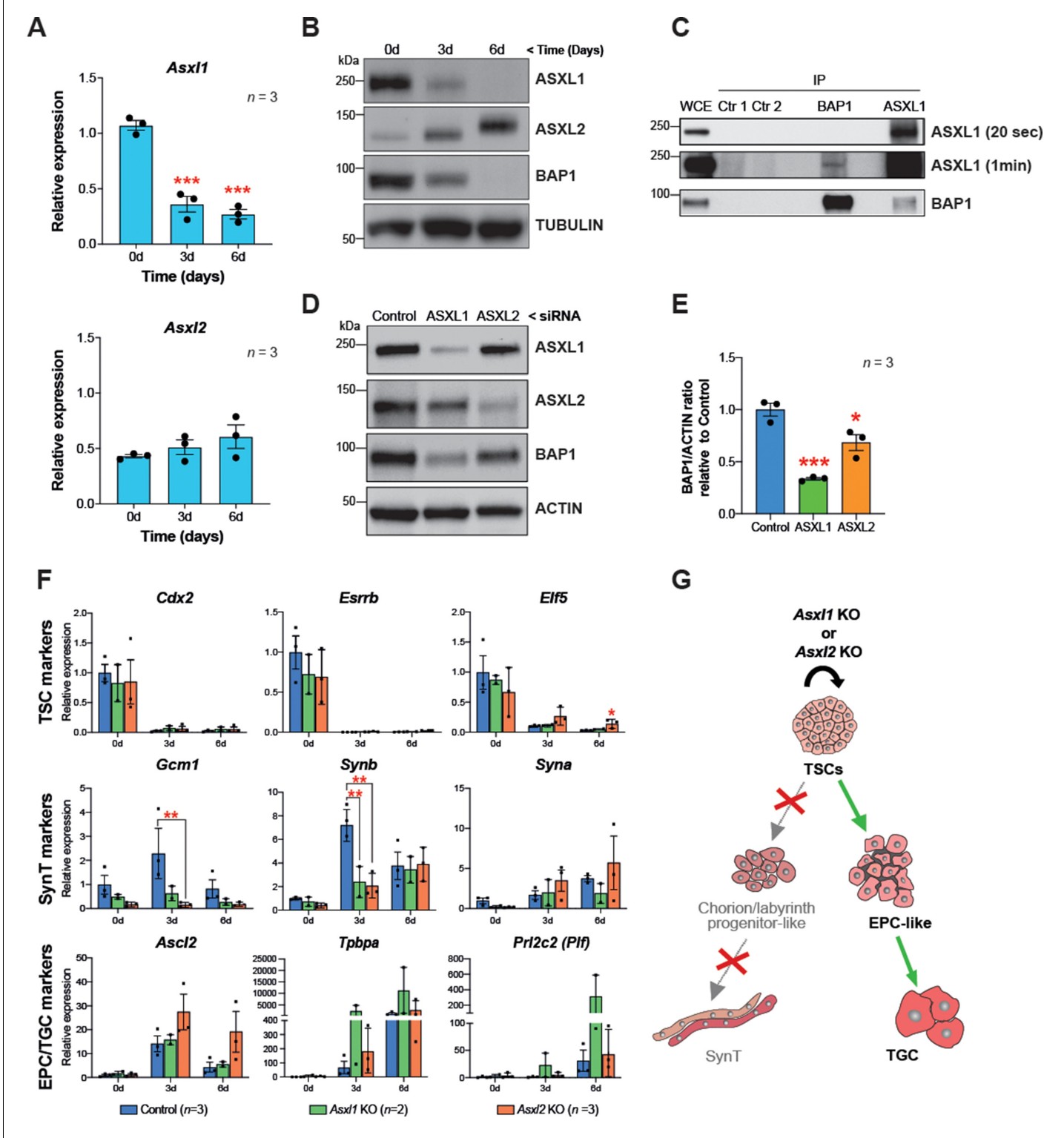

**Figure 5.** BAP1 and ASXL1/2 complexes are co-regulated during trophoblast differentiation. (**A**) RT-qPCR analysis of *Asxl1* and *Asxl2* expression during a 6-day differentiation time course of mouse trophoblast stem cells (mTSCs). Data are normalized to *Sdha* and are displayed as mean of three replicates ± SEM; ***p<0.001 (one-way ANOVA with Dunnett's multiple comparisons test). (**B**) Western blot analysis of ASXL1 and ASXL2 protein levels in mTSCs differentiating over 6 days (d). Blots shown are representative of three independent replicates. (**C**) Co-immunoprecipitation of endogenous BAP1 or ASXL1 proteins from mTSC whole cell extracts (WCE) (1 mg). WCE (20 μg) and immunoprecipitates (IP) were analysed by Western blot. Negative controls included protein A plus WCE (Ctr 1) and WCE plus protein A and isotype control Ab (Ctr 2). (**D**) siRNA-mediated knockdown of

*Figure 5 continued on next page*

*Figure 5 continued*

ASXL1 or ASXL2 followed by immunoblotting for the factors indicated. (**E**) Quantification of BAP1 levels (shown in D) normalized to ACTIN, displayed relative to the amounts in transfected control cells. Data are means ± SEM; n = 3. *p<0.05, ***p<0.001 (one-way ANOVA with Dunnett's multiple comparisons test). (**F**) Analysis of *Asxl1*$^{-/-}$ and *Asxl2*$^{-/-}$ mTSCs grown in self-renewal conditions (0d) or after 3d and 6d of differentiation assessed by RT-qPCR. Data are mean ± SEM of n = 3 (control, scramble), n = 2 (*Asxl1* KO), and n = 3 (*Asxl2* KO) individual clones as independent replicates. **p<0.01 (two-way ANOVA with Sidak's multiple comparisons test). (**G**) Schematic diagram of the differentiation defects observed in *Asxl1*$^{-/-}$ and *Asxl2*$^{-/-}$ mTSCs. The online version of this article includes the following figure supplement(s) for figure 5:

**Figure supplement 1.** CRISPR-mediated knockout (KO) of *Asxl1* and *Asxl2* in mouse trophoblast stem cells (mTSCs).

---

(*Chittock et al., 2017*), leading to the question whether BAP1 also functions to regulate trophoblast differentiation and invasion during human placentation.

To determine the dynamics of *BAP1* expression in human trophoblast, we first performed RT-qPCRs on placental villous biopsies across gestation. Despite some variability, *BAP1* mRNA levels increased over the course of pregnancy (*Figure 6A*). The expression of *BAP1* was also analysed in hTSCs and choriocarcinoma cell lines. Interestingly, among the placental choriocarcinoma cell lines, the most invasive cell line JEG-3 (*Grümmer et al., 1994*) showed lowest BAP1 expression levels compared to JAR and BeWo cell lines, suggesting that BAP1 may play a role in regulating trophoblast invasion also during placentation in humans (*Figure 6B*). In first trimester placentae, strong expression of BAP1 was observed in villous cytotrophoblast (VCT) and at the base of cytotrophoblast cell columns (CCC) compared to the very low signal in syncytiotrophoblast (*Figure 6C*). Of note, BAP1 staining became markedly weaker and more diffuse along the distal aspects of the CCC as cells undergo EMT and differentiate towards invasive EVT (*Figure 6C*). High expression of integrin alpha-5 (ITGA5), a marker of EVT, correlated with decreased staining intensity of BAP1, suggesting that BAP1 was downregulated during EVT differentiation (*Figure 6—figure supplement 1A*).

To further corroborate these results, we differentiated hTSCs towards EVT for 8 days (*Okae et al., 2018*) and examined the expression of BAP1/ASXL complex components by RT-qPCR. We confirmed successful EVT differentiation by *HLA-G* expression and concomitant downregulation of *CDH1*. Although *BAP1* mRNA expression levels remain unchanged, protein levels declined markedly upon EVT differentiation (*Figure 6D and E*), in line with the post-transcriptional regulation of BAP we had observed in the mouse (*Figure 1D and E*). In addition to *ASXL1* and *ASXL2*, the *ASXL3* family member was also expressed in hTSCs. Both *ASXL1* and *ASXL3* were significantly downregulated upon EVT differentiation (*Figure 6D*). To gain insight into the molecular function of BAP1 in human trophoblast, we performed overexpression experiments in hTSCs by lentiviral transduction of GFP-BAP1 compared to a GFP vector control plasmid (*Figure 6F and G*), and examined the expression of genes involved in stem cell self-renewal and EMT by WB and RT-qPCR. Whereas *BAP1* overexpression did not affect hTSC genes such as *GATA3*, *TEAD4*, or *ITGA2*, the cytotrophoblast cell transcription factor *ELF5* was upregulated (*Lee et al., 2018*; *Okae et al., 2018*; *Hemberger et al., 2010*). More significantly, however, the epithelial markers *CDH1*, *CLDN2*, *TJP1*, and *VCL* were significantly upregulated with a concomitant strong repression of mesenchymal marker genes *CDH2* and *ZEB2* (*Figure 6F and G* and *Figure 6—figure supplement 1B*). These data demonstrate that as in mouse, the levels of BAP1 are chief regulators of epithelial cell integrity and EMT progression in human trophoblast (*Figure 6G* and *Figure 6—figure supplement 1B*). Taken together, these results indicate that the molecular mechanism by which the BAP1/ASXL complexes regulate trophoblast differentiation and invasion may be conserved in human and in mice.

## Discussion

The similarities between trophoblast and tumour cells have long been recognized, in particular with respect to their invasive properties (*Costanzo et al., 2018*; *Ferretti et al., 2007*). BAP1, a tumour suppressor frequently mutated in human cancers, is ubiquitously expressed and inactivation or deletion of this gene results in metastasis (*Carbone et al., 2013*). During murine development, embryos deficient for *Bap1* die during around mid-gestation (E9.5) with severe placental dysmorphologies. Although a central role for BAP1 during early placentation was suggested (*Perez-Garcia et al., 2018*), its function in regulating trophoblast development has not been explored to date. In the current study, we show that BAP1 is highly expressed in both mTSCs and hTSCs, and that its

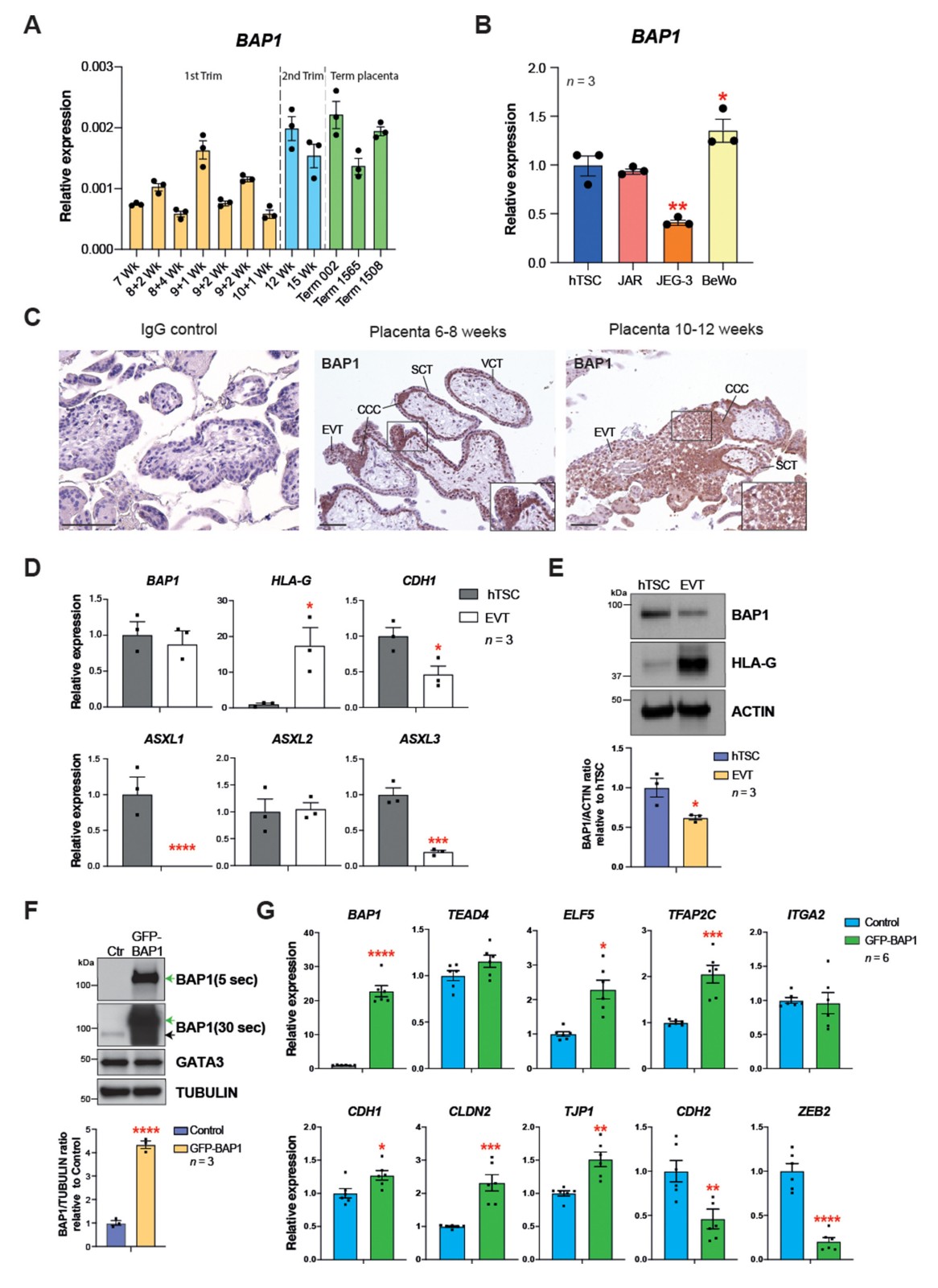

**Figure 6.** BRCA1-associated protein 1 (BAP1) polycomb repressive deubiquitinase (PR-DUB) modulation is also observed in human placentation. (**A**) RT–qPCR analysis of *BAP1* expression on human placental villous samples ranging from 7 weeks of gestation to term. Three independent term placental samples were investigated. An overall increase of *BAP1* expression was observed over gestation. Expression is normalized to *YWHAZ* housekeeping gene. Data are mean of three replicates ± SEM. (**B**) RT-qPCR analysis of *BAP1* expression in human trophoblast stem cells (hTSCs) and

*Figure 6 continued on next page*

*Figure 6 continued*

the choriocarcinoma cell lines JAR, JEG-3, and BeWo. Expression is normalized to *GAPDH*. Data are mean of three replicates ± SEM; *p<0.05, **p<0.01 (one-way ANOVA with Dunnett's multiple comparisons test). (C) Immunohistochemistry for BAP1 on early (6–8 weeks [wk] of gestation) and late first trimester placentae (10–12 weeks of gestation). BAP1 staining is strong in proliferative villous cytotrophoblast (VCT) and cytotrophoblast cell columns (CCC) compared to syncytiotrophoblast (SCT). Notably, invasive extravillous trophoblast (EVT) shows a diffuse and weak staining as cells undergo EMT. Representative images of three biological replicates. Scale bar: 100 μm. (D) RT-qPCR analysis of *BAP1, HLA-G, CDH1,* and *ASXL1-3* gene expression on hTSCs and in vitro-differentiated EVT cells after 8 days of differentiation. Expression is normalized to *GAPDH*. Data are mean of three independent replicates ± SEM; *p<0.05, ***p<0.001, ****p<0.0001 (Student's t-test). (E) Western blot analysis of BAP1 protein levels in EVT compared to hTSCs. As in the mouse, BAP1 is strongly downregulated during trophoblast differentiation towards the invasive EVT lineage. Graph shows the quantification of three independent replicates. Data are mean ± SEM; *p<0.05 (Student's t-test). (F) hTSCs transduced with GFP-empty control or GFP-BAP1 lentiviral particles were isolated by using fluorescence activated cell sorting (FACS), grown in stem cell conditions and examined by Western blotting. TUBULIN was used as loading control. Green arrows point to the exogenous GFP-BAP1 band after 5 and 30 seconds (sec) of film exposure. Black arrow points to endogenous BAP1. Graph shows the quantification of three independent replicates. Data are mean ± SEM; ****p<0.0001 (Student's t-test). (G) RT-qPCR analysis of control and GFP-BAP1-transduced hTSCs grown in stem cell conditions. Expression is normalized to *TBP* housekeeping gene expression. Data are mean of six independent replicates ± SEM; *p<0.05, **p<0.01, ***p<0.001, ****p<0.0001 (Student's t-test).

The online version of this article includes the following figure supplement(s) for figure 6:

**Figure supplement 1.** BAP1 immunofluorescence staining of first trimester human placenta.

downregulation triggers EMT and promotes trophoblast invasiveness. We also find that BAP1 protein levels are tightly coordinated with the expression of the ASXL proteins, indicating that modulation of the PR-DUB complex is required for proper trophoblast differentiation. To unravel the mechanism by which BAP1 and ASXL may regulate trophoblast self-renewal and differentiation, we deleted and overexpressed these factors in mTSCs using CRISPR/Cas9-KO and CRISPR/Cas9-SAM activation technology – the first time this method of gene expression modulation has been employed in mTSCs to date. We find that BAP1 regulates many facets of trophoblast biology. First, in stem cell conditions, *Bap1* ablation triggers an overt EMT phenotype associated with increased cellular invasiveness, and, at the same time, enhances proliferation and expression of stem cell markers. This suggests that deficiency of *Bap1* uncouples the normal loss of proliferation from differentiation, reminiscent of malignant transformation. Indeed, the upregulation of stem cell markers upon functional BAP1 depletion is seen in human uveal melanoma and renal cell carcinoma, and is associated with aggressive cancer behaviour and poor patient outcome (*Matatall et al., 2013*; *Peña-Llopis et al., 2012*; *Harbour et al., 2010*). In line with these data, overexpression of BAP1 induces the converse phenotype in mTSCs with reinforcement of epithelial features and reduced invasiveness. Therefore, we propose that BAP1 modulation, and specifically its downregulation, is one of the main drivers triggering the EMT and invasion processes in trophoblast. In line with this view, BAP1 mutations in human liver organoids result in loss of cell polarity, epithelial disruption, and increased cell motility, features observed during the initial steps of EMT (*Kalluri and Weinberg, 2009*; *Das et al., 2019*).

To the best of our knowledge, a direct link between BAP1 modulation and EMT regulation in early development and cancer has not been reported. However, results from previous studies of cancers that are strongly associated with an EMT process during metastatic transformation such as uveal melanoma, clear-cell renal cell carcinoma, gastric adenocarcinoma, colorectal cancer, and non-small-cell lung cancer showed a significant decrease in tumour BAP1 expression and worse clinical outcomes (*Kalirai et al., 2014*; *Yan et al., 2016*; *Tang et al., 2013*; *Fan et al., 2012*). On the background of our results reported here, it is tempting to speculate that the characteristic EMT and metastatic behaviour of these malignancies is induced by deletion or low abundance of BAP1. However, loss of BAP1 has also been reported to promote mesenchymal-epithelial transition in kidney tumours cells suggesting that its precise mode of function depends on the cell- and tissue-specific context (*Chen et al., 2019*). Further molecular analyses will be required to unravel the intricate regulation of the EMT pathway in different cellular contexts and the role of BAP1 in these processes.

In the absence of an FGF signal, *Bap1* deficiency promotes trophoblast differentiation towards a TGC phenotype, while syncytiotrophoblast formation is repressed. This is shown by the precocious cytoskeletal rearrangements we describe in mutant mTSCs, as well as the profound overabundance of terminally differentiated, extremely large TGCs in the KO placentae (*Perez-Garcia et al., 2018*). These dual roles of BAP1 depending on the FGF signalling environment may indeed be explained

by the differential regulation of its PR-DUB components, ASXL1 and ASXL2. BAP1 and ASXL proteins form mutually exclusive complexes of the PR-DUB tumour suppressor, which maintains transcriptional silencing of polycomb target genes. Moreover, the fact that ASXL proteins can affect the stability of BAP1 may account for the high discrepancy between mRNA and protein levels in m/hTSCs (*Scheuermann et al., 2010*; *Daou et al., 2015*). Our data suggest that the BAP1:ASXL1 complex is predominant in mTSCs and plays an important role in preventing mTSCs from undergoing EMT while in a proliferative stem cell state. With the onset of trophoblast differentiation and the concomitant upregulation of ASXL2, both BAP1:ASXL1 and BAP1:ASXL2 complexes are likely to coexist. Both these complexes are important to promote syncytiotrophoblast differentiation; in the absence of either *Bap1*, *Asxl1*, or *Asxl2*, syncytiotrophoblast differentiation is abrogated, and TGC differentiation dominates. This is in keeping with the finding that overexpression of *Asxl2* induces cellular senescence in other systems (*Huether et al., 2014*; *Micol et al., 2014*; *Daou et al., 2015*).

We previously found a strong correlation between cardiovascular and brain defects in embryos with abnormal placentation (*Woods et al., 2018*). Since *Asxl1* and *Asxl2* mutants have also been reported to exhibit cardiovascular and brain developmental defects (*Baskind et al., 2009*; *Wang et al., 2014*), it is tempting to speculate that they may be due, in part, to a placental defect.

Finally, our data suggest that BAP1:ASXL1/2 regulate trophoblast differentiation and invasiveness in other species. In humans as in mice, BAP1 protein levels are downregulated during differentiation towards the invasive EVT lineage in coordination with ASXL gene expression. We also observed that in addition to *ASXL1* downregulation, *ASXL3* expression was also modulated during EVT differentiation. De novo mutations of *ASXL1/2/3* genes are associated with severe FGR, preterm birth, and defects in the development of the heart-brain axis (*Srivastava et al., 2016*). These types of defects are strongly linked to abnormal placentation. Our work suggests a direct link of these mutations to abnormal trophoblast development through the various functions of PR-DUB in regulating the unique properties of trophoblast cells. Gaining detailed insights into the molecular networks regulating this BAP1-ASXL modulation during early placentation will help not only to shed light onto the major unexplained pregnancy disorders, but also to open up new avenues into investigations of tumours where PR-DUB is mutated.

# Materials and methods

**Key resources table**

| Reagent type (species) or resource | Designation | Source or reference | Identifiers | Additional information |
|---|---|---|---|---|
| Gene (*Mus musculus*) | *Bap1* | GenBank | MGI:1206586 | |
| Gene (*Mus musculus*) | *Asxl1* | GenBank | MGI:2684063 | |
| Gene (*Mus musculus*) | *Asxl2* | GenBank | MGI:1922552 | |
| Gene (*Homo sapiens*) | *BAP1* | GenBank | HGNC:950 | |
| Cell line (*Mus musculus*) | Mouse trophoblast stem cell line | Prof. Rossant lab | TS-Rs26 | |
| Cell line (*Homo sapiens*) | Human trophoblast stem cell line | Prof. Arima lab | BTS5 | |
| Cell line (*Homo sapiens*) | HEK293T: Human embryonic kidney cells | ATCC CRL-3216 | HEK293T | |
| Cell line (*Homo sapiens*) | Choriocarcinoma cell line | *ATCC* HTB-36 | JEG-3 | |
| Cell line (*Homo sapiens*) | Choriocarcinoma cell line | *ATCC* HTB-144 | JAR | |

*Continued on next page*

*Continued*

| Reagent type (species) or resource | Designation | Source or reference | Identifiers | Additional information |
|---|---|---|---|---|
| Cell line (*Homo sapiens*) | Choriocarcinoma cell line | *ATCC* CCL-98 | BeWo | |
| Antibody | Anti-ACTIN (mouse monoclonal) | Abcam | ab6276 | WB (1:5000) |
| Antibody | Anti-ASXL1 (rabbit monoclonal) | Cell Signaling | #52519 | WB (1:1000), IF (1:100) |
| Antibody | Anti-ASXL1 (mouse monoclonal) | Abnova | H00171023-M05 | WB (1:1000), IP (1:100) |
| Antibody | Anti-ASXL2 (rabbit polyclonal) | Abcam | ab106540 | WB (1:1000), IF (1:100) |
| Antibody | Anti-ASXL2 (rabbit polyclonal) | Bethyl Laboratories | A302-037A | WB (1:1000), IP (1:250) |
| Antibody | Anti-BAP1 (rabbit monoclonal) | Cell Signaling | #13187 | IF (1:200), WB (1:1000), IHC (1:100), IP (1:250) |
| Antibody | Anti-E-Cadherin (CDH1) (mouse monoclonal) | BD Biosciences | 610181 | IF (1:200) |
| Antibody | Anti-CDX2 (mouse monoclonal) | Biogenex | MU392A-UC | WB (1:1000) |
| Antibody | Anti-ESRRB (mouse monoclonal) | R&D Systems | H6707 | WB (1:1000), IF (1:200) |
| Antibody | Anti-GATA3 (mouse monoclonal) | Invitrogen | MA5-15387 | WB (1:1000) |
| Antibody | Anti-HLAG (mouse monoclonal) | Bio-Rad | MCA2043 | WB (1:1000) |
| Antibody | Anti-ITGA5 (mouse monoclonal) | Santa Cruz Biotechnology | sc-376199 | IF (1:100) |
| Antibody | Anti-TUBULIN (rat monoclonal) | Abcam | ab6160 | WB (1:5000) |
| Antibody | Anti-Rabbit IgG (H + L)-HRP (goat polyclonal) | Bio-Rad | 170–6515 | WB (1:5000) |
| Antibody | Anti-Mouse IgG (H + L)-HRP (goat polyclonal) | Bio-Rad | 170–6516 | WB (1:5000) |
| Antibody | Anti-Rabbit IgG (H + L), Alexa Fluor 488 (goat polyclonal) | Thermo Fisher Scientific | A-11034 | IF (1:500) |
| Antibody | Anti-Mouse IgG (H + L), Alexa Fluor 568 (donkey polyclonal) | Thermo Fisher Scientific | A-10037 | IF (1:500) |
| Recombinant DNA reagent | Lenti dCas9-VP64-Blast (plasmid) | Addgene | RRID:Addgene_61425 | Lentiviral plasmid to transfect HEK293T cells and package dCas9-VP64 viral particles |
| Recombinant DNA reagent | Lenti MS2-p65-HSF1-Hygro (plasmid) | Addgene | RRID:Addgene_61426 | Lentiviral plasmid to transfect HEK293T cells and package MS2-p65-HSF1 viral particles |

*Continued on next page*

*Continued*

| Reagent type (species) or resource | Designation | Source or reference | Identifiers | Additional information |
|---|---|---|---|---|
| Recombinant DNA reagent | Lenti sgRNA (MS2)-puro (plasmid) | Addgene | RRID:Addgene_73795 | Lentiviral plasmid to transfect HEK293T cells and package sgRNA viral particles |
| Recombinant DNA reagent | psPAX2 (plasmid) | Addgene | RRID:Addgene_12260 | Lentiviral packaging plasmid |
| Recombinant DNA reagent | pMD2.G (plasmid) | Addgene | RRID:Addgene_12259 | VSV-G envelope expressing plasmid |
| Recombinant DNA reagent | pSpCas9(BB)—2A-GFP (PX458) (plasmid) | Addgene | RRID:Addgene_48138 | Plasmid to express Cas9 from *Streptococcus pyogenes* with 2A-EGFP, and cloning backbone for CRISPR-knockout sgRNA |
| Recombinant DNA reagent | pLV [Exp]-Puro-CMV > EGFP:mBap1[NM_027088.2] (lentiviral particles) | VectorBuilder | Calves(VB210106-1179qkj)-C | Lentiviral particles to transduce and express mouse GFP-BAP1 |
| Recombinant DNA reagent | pLV [Exp]-Puro-CMV > EGFP:hBAP1[NM_004656.4] (lentiviral particles) | VectorBuilder | Cat#LVS(VB210106-1177amh)-C | Lentiviral particles to transduce and express human GFP-BAP1 |
| Sequence-based reagent | Stealth small interfering RNA (siRNA) | Thermo Fisher Scientific | 1320003_MS23-25 | |
| Sequence-based reagent | Stealth small interfering RNA (siRNA) | Thermo Fisher Scientific | 1320003_MS36-38 | |
| Sequence-based reagent | Stealth RNAi siRNA Negative Control Kit | Thermo Fisher Scientific | 12935100 | |
| Peptide, recombinant protein | Alexa Fluor 568 Phalloidin | Thermo Fisher Scientific | A12380 | IF (1:500) |
| Chemical compound, drug | Lipofectamine RNAiMAX Transfection Reagent | Thermo Fisher Scientific | 13778 | |
| Chemical compound, drug | Lipofectamine 2000 Transfection Reagent | Thermo Fisher Scientific | 11668019 | |
| Commercial assay or kit | Vybrant cell adhesion assay kit | Thermo Fisher Scientific | V13181 | |
| Software, algorithm | Fiji | *Schindelin et al., 2012* | | |
| Software, algorithm | StarDist | *Schmidt et al., 2018* | | |
| Software, algorithm | TrackMate | *Tinevez et al., 2017* | | |
| Software, algorithm | MorphoLibJ | *Legland et al., 2016* | | |
| Software, algorithm | R studio | R studio software | http://www.rstudio.com | |
| Software, algorithm | GraphPad Prism 8 | GraphPad software | http://www.graphpad.com | |

## Cell culture and generation of mutant TSC lines

The wild-type TS-Rs26 TSC line (a kind gift of the Rossant lab, Toronto, Canada) and mutant TSC lines were grown as previously described (*Tanaka et al., 1998*). Briefly, mTSCs were cultured in standard mTSC conditions: 20% fetal bovine serum (FBS) (Thermo Fisher Scientific 10270106), 1 mM sodium pyruvate (Thermo Fisher Scientific 11360–039), 1× anti-mycotic/antibiotic (Thermo Fisher Scientific 15240–062), 50 µM 2-mercaptoethanol (Gibco 31350), 37.5 ng/ml bFGF (Cambridge Stem Cell Institute), and 1 µg/ml heparin in RPMI 1640 with L-glutamine (Thermo Fisher Scientific 21875–034), with 70% of the medium pre-conditioned on mouse embryonic fibroblasts (CM). The medium was changed every 2 days, and cells passaged before reaching confluency. Trypsinization (0.25% trypsin/EDTA) was carried out at 37°C for about 5 min. Differentiation medium consisted of unconditioned TSC medium without bFGF and heparin.

*Bap1* KO mTSC clones were generated in our laboratory and published before (*Perez-Garcia et al., 2018*). *Bap1* was overexpressed in mTSCs by using CRISPR/Cas9 SAM system (*Konermann et al., 2015*). In brief, SAM mTSCs were generated by lentiviral transduction of lenti dCas9-VP64-Blast (Addgene 61425) and lenti MS2-p65-HSF1-Hygro (Addgene 61426) into TS-Rs26 mTSCs, followed by antibiotic selection. Then, to generate SAM *Bap1* mTSCs, three *Bap1*-gRNA targeting the 180 bp region upstream of the *Bap1* TSS and one non-targeting-gRNA (*Supplementary file 6*; *Joung et al., 2019*) were selected and synthesized (Sigma). Each oligo was annealed and cloned into the sgRNA (MS2)-puro plasmid (Addgene 73795) by a Golden Gate reaction using BsmBI enzyme (Thermo Fisher Scientific, ER0451) and T7 ligase (NEB, M0318S). The new gRNA constructs were packaged into lentiviral particles and transduced into SAM mTSCs by direct supplementation of the lentivirus for 24 hr. After 48 hr, SAM *Bap1*-overexpressing cells were selected by adding 1 µg/1 ml puromycin for 7 days.

CRISPR/Cas9-mediated *Asxl1*- and *Asxl2*-mutant mTSCs were generated as in *López-Tello et al., 2019*. Briefly, non-targeting gRNA (control) and gRNAs (*Supplementary file 6*) that result in frameshift mutations were designed using the CRISPR.mit.edu design software and cloned into the Cas9.2A.EGFP plasmid (Plasmid #48138 Addgene). Transfection of gRNA Cas9.2A.EGFP constructs was carried out with Lipofectamine 2000 (Thermo Fisher Scientific 11668019) reagent according to the manufacturer's protocol. KO clones were confirmed by genotyping using primers spanning the deleted exon, and by RT-qPCR with primers within, and downstream of, the deleted exon, as shown (*Figure 5—figure supplement 1C*).

BTS5 blastocyst-derived hTSCs were obtained from Prof. Takahiro Arima and cultured as in *Okae et al., 2018*. Briefly, hTSCs were grown in 5 mg/ml Col IV-coated six-well plates (Sigma C7521) with 2 ml of TS medium (DMEM/F12 [Invitrogen 31330]) supplemented with 0.1 m 2-mercaptoethanol (Gibco 31350), 0.2% FBS (Thermo Fisher Scientific 10270106), 100 µg/ml Primocin (Invivogen ant-pm1), 0.3% BSA (Sigma A8412), 1% ITS-X supplement (Gibco 51500–056), 1.5 mg/ml L-ascorbic acid (Sigma A4403), 50 ng/ml EGF (Peprotech AF-100–15), 2 mM CHIR99021 (R&D 4423), 0.5 mM A83-01 (Stem Cell Technologies 72024), 1 mM SB431542 (Tocris 1614), 0.8 mM VPA (Sigma P4543), and 5 mM Y27632 (Stem Cell Technologies 72304). The medium was changed every 2 days, and cells were dissociated with TrypLE (Gibco 12604–021) for 10–15 min at 37°C to passage them. EVT differentiation was achieved through a modification of a protocol described previously (*Okae et al., 2018*). hTSCs were cultured in pre-coated six-well plates (1 µg/ml Col IV) with 2 ml of EVT differentiation medium (EVTM: DMEM/F12, 0.1 mM 2-mercaptoethanol [Gibco 31350]), 100 µg/ml Primocin (Invivogen ant-pm1), 0.3% BSA (Sigma A8412), 1% ITS-X supplement (Gibco 51500–056), 2.5 µM Y27632 (Stem Cell Technologies 72304), 100 ng/ml NRG1 (Cell Signaling 5218SC), 7.5 µM A83-01 (Tocris Biotechne 2939), and 4% knockout serum replacement (KSR) (Thermo Fisher 10828010). Matrigel (Corning 356231) at 2% final concentration was added as cells were suspended in medium and seeded in the plate. After 3 days, EVTM was changed and replaced with new EVTM without NRG1 and a final Matrigel concentration of 0.5%. At 6 days of differentiation, EVTM was replaced with new EVTM without NRG1 and KSR. EVTs were cultured 2 more days and then collected for RNA and protein extraction.

The choriocarcinoma cell lines JEG-3 and JAR were cultured in RMPI-1640 (Thermo Fisher Scientific, Waltham, MA) supplemented with 10% (v/v) FBS, 2 mM glutamine, penicillin (10 U/ml), streptomycin (100 µg/ml), and gentamicin (2 mg/ml) (Thermo Fisher Scientific, Waltham, MA). BeWo cells

were cultured in DMEM/F12 medium supplemented with 10% HI-FBS, penicillin (100 U/ml), and streptomycin (100 µg/ml) (Thermo Fisher Scientific, Waltham, MA). Culture medium was replaced every 2–3 days. Approximately 4–6 days after plating, cells were removed from tissue culture flasks with TrypLE (Gibco) to be either passaged at a ratio of 1:3 or collected for RNA extraction and RT-qPCR analysis. All cell lines used were proven mycoplasma-free.

## Human samples

The placental samples from normal first and early second trimester, and normal term pregnancies used for this study were obtained with written informed consent from all participants in accordance with the guidelines in the Declaration of Helsinki 2000. Elective terminations of normal pregnancies were performed at Addenbrooke's Hospital under ethical approval from the Cambridge Local Research Ethics Committee (04/Q0108/23). Samples were either snap-frozen for RNA isolation or embedded in formalin-fixed paraffin wax for tissue sections (4 µm).

## Mice

All animal experiments were conducted in full compliance with UK Home Office regulations (Animals Act 1986) and with approval of the local animal welfare committee (AWERB) at the Babraham Institute, and with the relevant project and personal licences in place. All conceptuses used in this study were dissected at E6.5 from C57BL/6Babr mice bred and maintained in the Babraham Institute Biological Support Unit.

## Lentiviral transduction

For the production of lentiviral particles, 106 HEK293T cells seeded in 100 mm plates were cotransfected (TransIT, Mirus BIO 2700) with 6.5 µg of psPAX2 (Addgene 12260), 3.5 µg of pMD2.G (Addgene 12259), and 10 µg of the appropriate lentiviral vector: dCas9-VP64_Blast (Addgene 61425), MS2-p65-HSF1_Hygro (Addgene 61426), sgRNA(MS2)-puro (Addgene 73795) cloned with an individual sgRNA. Forty-eight hours later, 10 ml of virus supernatant was filtered through a 0.45 µm filter (Sartorius, 16533) and supplemented with 8 µg/ml polybrene (Millipore, TR-1003-G). For the specific case of mouse and human GFP-BAP1 overexpression experiments, lentiviral particles were packaged by VectorBuilder and supplemented to the specific cell lines as before.

Small interference RNA transfection mTSCs were transfected with Stealth small interfering RNA (siRNA) for *Asxl1, Asxl2,* and negative control siRNA (Thermo Fisher Scientific, 1320003MS23-25, 1320003MS36-38, and 12935100, respectively) using Lipofectamine RNAiMax (Thermo Fisher Scientific, 13778075). Following 72 hr of transfection, cells were collected for WB analysis.

## WB and immunoprecipitation

Cells were lysed in radioimmunoprecipitation assay buffer (20 mM Tris-HCl, pH 8.0, 137 mM NaCl, 1 mM MgCl$_2$, 1 mM CaCl$_2$, 10% glycerol, 1% NP-40, 0.5% sodium deoxycholate, 0.1% sodium dodecyl sulphate), containing a protease inhibitor cocktail (Sigma P2714), and incubated at 4°C for 1 hr, followed by centrifugation (9300× *g*, 10 min). Western blotting was performed as described previously (*Pérez-García et al., 2014*). Blots were probe against the antibodies anti-Bap11:1000 (Cell Signaling, D1W9B #13187), anti-beta-ACTIN 1:5000 (Abcam, ab6276), anti-TUBULIN 1:5000 (Abcam, ab6160), anti-CDX2 1:1000 (Biogenex, MU392A-UC), anti-ESRRB 1:1000 (R&D Systems, H6707), Anti-ASXl1 1:1000 (Cell Signaling, D1B6V #52519), Anti-ASXl1 1:1000 (Abnova, H00171023-M05), anti-ASXl2 1:1000 (Abcam, ab106540), anti-ASXl2 1:1000 (Bethyl Laboratories, MCA2043), anti-GATA3 1:1000 (Invitrogen, MA5-15387), and anti-HLA-G (Bio-Rad, MCA2043) followed by horseradish peroxidise-conjugated secondary antibodies anti-rabbit (Bio-Rad 170–6515), anti-mouse (Bio-Rad 170–6516, all 1:3000). Detection was carried out with enhanced chemiluminescence reaction (GE Healthcare RPN2209) on X-ray films. The intensity of the bands was quantified using ImageJ software.

For optimal detection of the BAP1 and ASXL co-immunoprecipitation, cells were lysed in a detergent buffer (10 mM Tris-HCl [pH 7.4], 150 mM NaCl, 10 mM KCl, 0.5% Nonidet P-40) with a protease inhibitor cocktail (Sigma P2714) and incubated at 4°C for 1 hr, followed by centrifugation (9300× *g*, 10 min). The immunoprecipitation was performed as described previously (*Pérez-García et al., 2014*).

## Immunohistochemistry

Immunohistochemistry on sections of E9.5 wild-type and *Bap1* KO placentas from the DMDD collection (dmdd.org.uk) and first trimester placentas was performed as in *Turco et al., 2018*. Briefly, immunohistochemistry was carried out using heat-induced epitope retrieval buffers (A. Menarini) and Vectastain avidin-biotin-HRP reagents (Vector Laboratories PK-6100). Anti-BAP1 antibody (Cell Signaling, D1W9B #13187) was used at 1:200. For each experiment, a negative control was included in which the antibody was replaced with equivalent concentrations of isotype-matched rabbit IgG. Images were taken with an EVOS M5000 microscope (Thermo Fisher Scientific).

## Immunofluorescence staining

Cells were fixed with 4% paraformaldehyde (PFA) in phosphate-buffered saline (PBS) for 10 min and permeabilized with PBS, 0.3% Triton X-100 for 10 min. Blocking was carried out with PBS, 0.1% Tween 20, 1% BSA (PBT/BSA) for 30 min, followed by antibody incubation for 60 min. Primary antibodies and dilutions (in PBT/BSA) were E-cadherin (CDH1) 1:200 (BD Biosciences, 610181), anti-BAP1 1:200 (Cell Signaling, D1W9B #13187), anti-ESRRB 1:200 (R&D Systems H6707), and phalloidin 1:500 (Thermo Fisher Scientific A12380).

Cryosections of placental villi from a total of three samples of 8 weeks of gestation were cut at 15 µm and fixed with ice-cold methanol/acetone for 10 min. Tissues were blocked with PBS, 0.5% bovine serum albumin (Sigma, A7906), 0.1% Tween-20. Antibodies and dilutions were: BAP1 1:100 (Cell Signaling, D1W9B #13187) and Anti-ITGA5 1:100 (Santa Cruz Biotechnology, sc-376199). Incubation was done at 4°C overnight. Detection was carried out with Alexa fluorophore-conjugated secondary antibodies (Thermo Fisher Scientific) diluted 1:400. Nuclear counterstaining was performed with DAPI.

Whole-mount embryo staining was performed following a modification of the protocol previously described (*Kalkan et al., 2019*). Briefly, dissected E6.5 conceptuses were fixed for 1 hr in 4% PFA. After three washes (15 min) with PBS supplemented with 3 mg/ml poly-vinylpyrrolidone (Sigma, P0930), embryos were permeabilized with PBS containing 5% DMSO, 0.5% Triton X-100, 0.1% BSA, 0.01% Tween 20 for 1 hr. Then, embryos were blocked overnight at 4°C in permeabilization buffer, containing 2% donkey serum. Embryos were incubated overnight at 4°C with antibodies against E-cadherin (CDH1) at 1:200 (BD Biosciences, 610181) and BAP1 at 1:100 (Cell Signaling, D1W9B #13187) in blocking buffer, followed by three washes in blocking buffer for 1 hr. Then, conceptuses were incubated overnight with secondary Alexa Fluor 488 or 568 (Thermo Fisher Scientific) antibodies diluted 1:400 in blocking buffer. Lastly, embryos were washed three times for 1 hr in blocking buffer and nuclei were counter-stained with DAPI. For embryo mounting, samples were taken through a series of 25%, 50%, 75%, and 100% Vectashield (Vector Laboratories, H-1000) diluted in PBS. Embryos were mounted in Vectashield, surrounded by spacer drops of vaseline for the coverslip, to immobilize embryos. Images were taken with an Olympus BX61 epifluorescence microscope or a Zeiss LSM 780 confocal microscope. Images were processed and analysed with a custom analysis pipeline developed in Fiji (*Schindelin et al., 2012*). In brief, first nuclei in 3D image stacks were segmented using a combination of StarDist (*Schmidt et al., 2018*) and TrackMate (*Tinevez et al., 2017*) plugins, followed by manual correction where required. This was followed by analysis of the morphology and the fluorescence intensity in all three channels (DAPI, CDH1, and BAP1) within all segmented nuclei using the MorpholibJ plugin (*Legland et al., 2016*). In addition, for all nuclei it was recorded whether they localized to the ExE or the EPC by creating 3D segmentation masks of the embryo, relying on the DAPI and CDH1 labelling. Post-processing and plotting of the data was done in R studio (http://www.rstudio.com). Student's t-tests analysis was performed to calculate statistical significance of BAP1 staining differences (p<0.05) using GraphPad Prism 8.

## Cell adhesion assay

Adhesion capacity of vector control and *Bap1*[-/-] mTSCs was measured using the Vybrant cell adhesion assay kit (Thermo Fisher Scientific V13181) as previously described (*Branco et al., 2016*). Briefly, cells resuspended in serum-free RPMI medium were labelled with Calcein AM (5 µM) during 30 min at 37°C. Cells were washed twice with RPMI medium and $10^5$ cells plated per well in a 96-well tissue culture plate and left to attach for 2 hr in serum-free RPMI medium. Finally, cells were washed three

times and the remaining attached cells were detected measuring the fluorescence emission at 517 nm with a PHERAstar FS plate reader.

## Trophosphere generation

Trophospheres were generated following a modification of a protocol described previously (**Rai and Cross, 2015**). In brief, $10^4$ wild-type and mutant cells resuspended in complete medium were cultured in Ultra-Low Attachment plates (Corning, Steuben County, NY). Forty-eight hours later, cells were collected, washed with PBS, and transferred back to Ultra-Low Attachment dishes with differentiation medium for another 7 days. Then, the trophospheres were collected for RNA analysis.

## Trophoblast cell invasion

The invasion assays were carried out following a modification of the protocol described in **Hemberger et al., 2004**. The Transwell filters (Sigma, CLS3422) were coated with 100 µl of a 1:20 dilution of cold Matrigel (Corning 356231) in RPMI 1640 medium. The Matrigel layer was allowed to dry overnight at room temperature and was rehydrated the next day with 100 µl of supplemented RPMI 1640 medium for 2 hr at 37°C under 95% humidity and 5% $CO_2$. Confluent 60 mm dishes of TS cells were trypsinized and resuspended in RPMI at $10^6$ cells/ml; 100 µl of this cell suspension ($10^5$ cells) was added to the top chamber, and the bottom chamber was filled with 800 µl of culture medium.

After the specific times of incubation, Transwell inserts were fixed for 5 min in 4% PFA and washed with 1× PBS. Cells that remained on top of the filters as well as the Matrigel coating were scraped off. Filters were stained overnight with hematoxylin and excised under a dissecting microscope, removing all residual cells from the top of the filters. Filters were mounted with 20% glycerol in 1× PBS and photographs of each filter were quantified with ImageJ.

## Proliferation assay

Analysis of cell proliferation rate was performed as in **Woods et al., 2017**. In brief, 10,000 vector control and *Bap1* KO mTSCs were plated in complete medium and collected every 24 hr over 4 days. After trypsinization, the number of viable cells was counted using the Muse Count and Viability Assay Kit (Merck Millipore MCH100102) and run on the Muse cell analyser (Merck Millipore), according to manufacturer's instructions. Statistical analysis was performed using ANOVA followed by Holm-Sidak's post hoc test.

## RT-qPCR

Total RNA was extracted using TRI reagent (Sigma T9424), DNase-treated, and 1 µg used for cDNA synthesis with RevertAid H-Minus reverse transcriptase (Thermo Fisher Scientific EP0451). Quantitative (q)PCR was performed using SYBR Green Jump Start Taq Ready Mix (Sigma S4438) and Intron-spanning primer pairs (**Supplementary file 6**) on a Bio-Rad CFX384 thermocycler. Normalized expression levels are displayed as mean relative to the vector control sample; error bars indicate standard error of the means (SEM) of at least three replicates. Where appropriate, Student's t-tests or ANOVA were performed to calculate statistical significance of expression differences ($p < 0.05$) using GraphPad Prism 8.

## RNA-seq

For RNA-seq, total RNA was extracted with Trizol followed by DNase treatment using TURBO DNA-free kit (Life Technologies AM1907). For wild-type and *Bap1* KO mTSC experiments, adapter indexed strand-specific RNA-seq libraries were generated from 1000 ng of total RNA following the dUTP method using the stranded mRNA LT sample kit (Illumina). Libraries were pooled and sequenced on Illumina HiSeq 2500 in 75 bp paired-end mode. FASTQ files were aligned to the *Mus musculus* GRCm38 genome reference genome using HISAT2 v2.1.0. Sequence data were deposited in ENA under accession ERP023265.

For RNA-seq from SAM *Bap1*-overexpressing cells, RNA-seq libraries were generated from 500 ng using TruSeq Stranded mRNA library prep (Illumina, 20020594). Indexed libraries were pooled and sequenced on an Illumina HiSeq2500 sequencer in 100 bp single-end mode. FastQ data were map to *M. musculus* GRCm38 genome assembly using HISAT2 v2.1.0.

## Bioinformatic analysis

Data were quantified using the RNA-seq quantitation pipeline in SeqMonk (http://www.bioinformat-ics.babraham.ac.uk) and normalized according to total read count (read per million mapped reads). Differential expression was calculated using DESeq2 and FPKM Fold Change $\geq$ 2 with p<0.05 and adjusted for multiple testing correction using the Benjamini-Hochberg method. Stringent differential expression was calculated combining DESeq2 and intensity difference filters in SeqMonk. Expression data from *Bap1*-mutant tumor cells were from (*He et al., 2019*).

Heatmaps and PCA plots were generated using Seqmonk. Gene ontology was performed on genes found to be significantly up- or downregulated, against a background list of genes consisting of those with more than 10 reads aligned. Gene ontology terms with a Bonferroni p-value of <0.05 were found using DAVID (*Dennis et al., 2003*). Venn diagrams were plotted using BioVenn (http://www.cmbi.ru.nl/cdd/biovenn/). The significance of the pairwise overlap between the datasets was determined by the overlapping gene group tool provided at http://www.nemates.org.

## Acknowledgements

We would like to thank Dr Anne Segonds-Pichon for expert help with statistical analyses, the Flow Cytometry Facility at the Babraham Institute and Kristina Tabbada for Illumina high-throughput sequencing.

## Additional information

### Funding

| Funder | Grant reference number | Author |
| --- | --- | --- |
| Wellcome Trust | WT100160MA | Myriam Hemberger |
| University of Cambridge | Next Generation Fellowship | Vicente Perez-Garcia |
| University of Cambridge | Centre for Trophoblast Research | Vicente Perez-Garcia |
| Spanish Ministry of Science and Innovation | RYC-2019-026956 | Vicente Perez-Garcia |
| European Commission | Erasmus+ traineeship | Pablo Lopez-Jimenez |
| Canadian Institutes of Health Research | Tier I Canada Research Chair | Myriam Hemberger |
| Canadian Institutes of Health Research | RN435448 - 450828 | Myriam Hemberger |

The funders had no role in study design, data collection and interpretation, or the decision to submit the work for publication.

### Author contributions

Vicente Perez-Garcia, Conceptualization, Data curation, Formal analysis, Funding acquisition, Investigation, Methodology, Writing - original draft, Project administration, Writing - review and editing; Georgia Lea, Pablo Lopez-Jimenez, Data curation, Formal analysis, Investigation; Hanneke Okkenhaug, Data curation, Formal analysis; Graham J Burton, Ashley Moffett, Margherita Y Turco, Writing - review and editing; Myriam Hemberger, Conceptualization, Supervision, Funding acquisition, Project administration, Writing - review and editing

### Author ORCIDs

Vicente Perez-Garcia (iD) https://orcid.org/0000-0001-5594-1607
Pablo Lopez-Jimenez (iD) https://orcid.org/0000-0002-6673-5996
Hanneke Okkenhaug (iD) https://orcid.org/0000-0003-0669-4069
Myriam Hemberger (iD) https://orcid.org/0000-0003-3332-6958

## Ethics

Human subjects: The placental samples from normal first and early second trimester, and normal term pregnancies used for this study were obtained, with written informed consent from all participants, in accordance with the guidelines in The Declaration of Helsinki 2000. Elective terminations of normal pregnancies were performed at Addenbrooke's Hospital under ethical approval from the Cambridge Local Research Ethics Committee (04/Q0108/23).

Animal experimentation: All animal experiments were conducted in full compliance with UK Home Office regulations (Animals Act 1986) and with approval of the local animal welfare committee (AWERB) at The Babraham Institute, and with the relevant project and personal licences in place. All conceptuses used in this study were dissected at E6.5 from C57BL/6Babr mice bred and maintained in the Babraham Institute Biological Support Unit.

## Decision letter and Author response

Decision letter https://doi.org/10.7554/eLife.63254.sa1
Author response https://doi.org/10.7554/eLife.63254.sa2

# Additional files

## Supplementary files

• Supplementary file 1. Genes dysregulated in Bap1 knockout (KO) mouse trophoblast stem cell (mTSC) in stem cell conditions (0 day) and during differentiation (3 days).

• Supplementary file 2. Gene ontology analyses of genes differentially expressed between vector and *Bap1*-mutant mouse trophoblast stem cells (mTSCs) in stem cell conditions (0 day) and during differentiation (3 days).

• Supplementary file 3. Genes dysregulated in Bap1 overexpressing mouse trophoblast stem cell (mTSC) in stem cell conditions (0 day) and during differentiation (3 days).

• Supplementary file 4. Gene ontology analyses of genes differentially expressed between Bap1 overexpressing and control mouse trophoblast stem cells (mTSCs) in stem cell conditions (0 day) and during differentiation (3 days).

• Supplementary file 5. List of genes commonly deregulated in the absence or excess of BRCA1-associated protein 1 (BAP1) in mouse trophoblast stem cells (mTSCs) growing in stem cell conditions and its gene ontology analyses.

• Supplementary file 6. Primer sequences for RT-qPCR and CRISPR gRNAs target sequences.

• Transparent reporting form

## Data availability

All data generated or analysed during this study are included in the manuscript and supporting files. Genome-wide sequencing data have been deposited in the GEO database under accession number GSE158670.

The following dataset was generated:

| Author(s) | Year | Dataset title | Dataset URL | Database and Identifier |
|---|---|---|---|---|
| Perez-Garcia V, Lea G, Lopez-Jimenez P, Okkenhaug H, Burton GJ, Moffett A, Turco MY, Hemberger M | 2020 | BAP1/ASXL complex modulation regulates Epithelial-Mesenchymal Transition during trophoblast differentiation and invasion | https://www.ncbi.nlm.nih.gov/geo/query/acc.cgi?acc=GSE158670 | NCBI Gene Expression Omnibus, GSE158670 |

The following previously published datasets were used:

| Author(s) | Year | Dataset title | Dataset URL | Database and Identifier |
|---|---|---|---|---|
| Perez-Garcia V, Fineberg E, Wilson R, Mazzeo AMCI, Tudor C, Sienerth A, White JK, Tuck E, Ryder EJ, Gleeson D, Siragher E, Wardle-Jones H, Staudt N, Wali N, Collins J, Geyer S, Busch-Nentwich EM, Galli A, Smith JC, Robertson E, Adams DJ, Weninger WJ, Mohun T, Hemberger M | 2018 | Placentation defects are highly prevalent in embryonic lethal mouse mutants | http://www.ebi.ac.uk/ena/data/view/ERP023265 | ENA, ERP023265 |
| He M, Chaurushiya MS, Webster JD, Kummerfeld S, Reja R, Chaudhuri S, Chen Y-J, Modrusan Z, Haley B, Dugger DL, Eastham-Anderson J, Lau S, Dey A, Caothien R, Roose-Girma M, Newton K, Dixit VM | 2019 | Intrinsic apoptosis shapes the tumor spectrum linked to inactivation of the deubiquitinase BAP1 | https://www.ncbi.nlm.nih.gov/geo/query/acc.cgi?acc=GSE120414 | NCBI Gene Expression Omnibus, GSE120414 |
| He M, Chaurushiya MS, Webster JD, Kummerfeld S, Reja R, Chaudhuri S, Chen Y-J, Modrusan Z, Haley B, Dugger DL, Eastham-Anderson J, Lau S, Dey A, Caothien R, Roose-Girma M, Newton K, Dixit VM | 2019 | Intrinsic apoptosis shapes the tumor spectrum linked to inactivation of the deubiquitinase BAP1 | https://www.ncbi.nlm.nih.gov/geo/query/acc.cgi?acc=GSE120415 | NCBI Gene Expression Omnibus, GSE120415 |

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
