## [Decision Letter]

**Acceptance summary:**

During implantation embryonic trophoblast cells invade into the maternal endometrium to establish the maternal-fetal interface. This work significantly advances our understanding of trophoblast differentiation by demonstrating that in the mouse Bap1/Asx1/2 complexes regulate epithelial to mesenchymal transition and invasive properties during this process. Experiments in human trophoblast stem cells suggest that this function of BAP1 is conserved in human.

**Decision letter after peer review:**

Thank you for submitting your article "BAP1/ASXL complex modulation regulates Epithelial-Mesenchymal Transition during trophoblast differentiation and invasion" for consideration by *eLife*. Your article has been reviewed by 3 peer reviewers, including Lilianna Solnica-Krezel as the Reviewing Editor and Reviewer #1, and the evaluation has been overseen Edward Morrisey as the Senior Editor.

The reviewers have discussed the reviews with one another and the Reviewing Editor has drafted this decision to help you prepare a revised submission.

Summary:

The manuscript from Perez-Garcia et al., follows up on a prior study by the same authors in which they identified the tumor suppressor BAP1 as a regulator of mouse placentation and trophoblast stem cells (TSCs) (Perez-Garcia et al., Nature, 2018). In their preceding work the authors showed that CRISPR-mediated knockout of BAP1 in TSCs results in upregulation of key stem cell markers Cdx2 and Esrrb and biased differentiation towards trophoblast giant cells at the expense of the syncytiotrophoblast lineage. Here the authors have expanded on these observations by demonstrating that BAP1 modulates the epithelial-to-mesenchymal transition (EMT) in TSCs and that a similar phenotype can be obtained by genetic deletion of Asxl1/2. Declining protein levels of BAP1 during differentiation of human TSCs into extravillous trophoblast suggest that the role of BAP1 may be conserved in humans. As the molecular mechanisms of trophoblast development, including EMT and invasive behaviors of trophoblast giant cells in the mouse and extravillous trophoblast cells in human are only beginning to be understood, this study provides an important advance.

This is a well-written and technically sound study that clarifies the role of BAP1 in trophoblast development. Overall, the work presented is very important to the fields of EMT and trophoblast stem cell biology, and it warrants publication in *eLife* in principle. However, the claims in the abstract, the model in Figure 5, and the conclusions in the discussion are not well-supported. Therefore, additional experimental work will be essential for the manuscript to become suitable for publication in *eLife*.

Essential revisions:

1. There was consensus that the current manuscript lacks functional data to demonstrate conservation of BAP1/ASXL1/2 function in human TSCs. These are crucial claims in the abstract that are not supported, and some elements of these claims are necessary for the manuscript to have impact beyond the previous Nature 2018 publication. Currently, the studies in human TSCs are purely observational (Figure 6D-E). The authors should employ genetic approaches to interrogate whether the functions of BAP1 in TSC self-renewal and differentiation are truly conserved between mouse and human.

2. The main takeaway from Figure 2 is that BAP1 is dispensable for mouse TSC maintenance and that BAP1 knockout results in increased expression of stem cell markers Cdx2 and Esrrb. Both of these findings were previously reported in the authors' 2018 paper (see Figure 4b in the Nature paper). Therefore, the statement that "BAP1 deletion does not impair the stem cell gene regulatory network" is not surprising and the authors should state clearly that these experiments confirm their prior observations.

3. The overexpression data in Figure 4 is difficult to interpret. Vector transduced TSCs show a tight, epithelial morphology (Figure 3A), whereas the NT-sgRNA control cells look like they are undergoing EMT (Figure 4C). Why does the introduction of the NT-sgRNA induce EMT characteristics? Bap1 sgRNA1 cells seem less epithelial than the Vector transduced cells. Do NT-sgRNA TSCs have less BAP1 than Vector transduced TSCs?

4. Moreover, all the data in Figure 4 are based on a single sgRNA that could activate BAP1 expression. To exclude off target effects, the authors should confirm the effect of BAP1 overexpression using another sgRNA or cDNA overexpression system.

5. The authors need to examine the gene expression data more closely as well as the functional consequences of BAP1 overexpression on TSC proliferation and differentiation. In particular it would be important to compare the list of DEG in BAP1 KO and overexpression condition. Are they mirror-image or are there differences? For example, *Zeb2* expression is strongly upregulated in BAP1 mutant line but not significantly altered in cells overexpressing BAP1. This should be discussed.

6. In the abstract, the authors state that BAP1 function during trophoblast development is dependent on its binding to Asxl1/2/3. However, the data presented in this manuscript do not address whether BAP1 and Asxl1/2/3 are indeed part of the same complex in TSCs. Furthermore, the fact that Asxl1/2 KO increases expression of syncytial genes (Figure 5) does not provide direct evidence of functional synergy between these proteins and BAP1. This conclusion could be strengthened by demonstrating that Asxl1 and BAP1 indeed have a protein-protein interaction in TSCs and/or by deleting the BAP1 binding domain in Asxl1/2. It would also be instructive to examine whether the phenotype of BAP1 overexpression in TSCs (e.g. gain of epithelial features and reduced invasiveness) is dependent on Asxl1. This could be examined by overexpressing BAP1 in Asxl1-deficient TSCs.

7. In some cases, experiments are carried out to "confirm" and "corroborate" hypotheses rather than test them. For example, the similarity between the gene expression signature of Bap1 mutant murine TSCs is and Bap1 mutant melanocytes and mesothelial cells is shown and emphasized. One wonders how unique is this similarity? Is Bap1 expression modulation observed in other EMT processes during development or in cancer? This should be explored and discussed.

[Editors' note: further revisions were suggested prior to acceptance, as described below.]

Thank you for resubmitting your work entitled "BAP1/ASXL complex modulation regulates Epithelial-Mesenchymal Transition during trophoblast differentiation and invasion" for further consideration by *eLife*. Your revised article has been evaluated by Edward Morrisey (Senior Editor) and a Reviewing Editor.

The manuscript has been improved and the reviewers are supportive of its publication in *eLife*. However, there are some remaining issues that need to be addressed, as outlined below:

The authors have substantially revised their manuscript in response to the reviewers' comments. This work significantly advances our understanding of trophoblast differentiation and in particular of Bap1/Asx1/2 complexes in regulation of EMT and invasive properties during this process, and should be of interest to the broad *eLife* readership. However, some conclusions should be reconsidered and the abstract clarity could be improved.

One of the key concerns to the original manuscript was that the conclusion " the molecular mechanism by which BAP1 regulates the epithelial characteristics of trophoblast is conserved between mice and human." was not supported by experimental data. In the revised manuscript new data are presented that over expression of BAP1 by viral transduction in a new hTSC line promotes epithelial characteristics. The authors argue that additional loss-of-function experiments for human BAP1 gene go beyond the scope of the manuscript. Whereas this reviewer agrees, the current data supports but does not "show that the molecular function of BAP1 is conserved in mouse and humans.", as is stated in the Abstract. This conclusion should be toned down in the abstract. In fact the language used at the end of Introduction more appropriately aligns with the current experimental results (lines 128-131) The functional characterization of BAP1 in the human placenta and human trophoblast stem cells (hTSCs) suggests that the role of BAP1 in regulating trophoblast differentiation and EMT progression is conserved in mice and humans. "

The clarity of the Abstract could be improved. In the sentence "Moreover, we show that this function is dependent on the binding of BAP1 to additional sex comb-like (ASXL1/2) proteins to form the Polycomb repressive deubiquitinase (PR-DUB) complex." it is not clear what "function" do the authors mean, as the previous sentence describes the effects of BAP1 protein downregulation from which BAP1 function (limiting EMT) needs to be deduced.

Line 367 "More significantly, however, the EMT markers CDH1, CLDN2" – CDH1 is not a key EMT marker, just the opposite. Please revise.

---

## [Author Response]

Essential revisions:1. There was consensus that the current manuscript lacks functional data to demonstrate conservation of BAP1/ASXL1/2 function in human TSCs. These are crucial claims in the abstract that are not supported, and some elements of these claims are necessary for the manuscript to have impact beyond the previous Nature 2018 publication. Currently, the studies in human TSCs are purely observational (Figure 6D-E). The authors should employ genetic approaches to interrogate whether the functions of BAP1 in TSC self-renewal and differentiation are truly conserved between mouse and human.

We agree with the reviewers’ statement that a more detailed investigation of BAP1 function in hTSCs will boost the scientific impact of this manuscript tremendously. To address this point, we have performed important additional functional experiments: Notably, we have generated BAP1 overexpressing hTSCs by lentiviral transduction followed by RT-qPCR to demonstrate that overexpression of BAP1 in hTSCs regulates the epithelial features of human trophoblast. As in mice, the overexpression of BAP1 in hTSCs did not affect their self-renewal capacity but resulted in upregulation of key EMT markers with a concomitant strong repression of EMT drivers. Thus, mesenchymal markers *CDH2* and *ZEB2* were markedly downregulated in BAP1-overexpressing cells, whereas pro-epithelial genes such as *CDH1*, *CLDN2* and *TJP1* were upregulated. These data are in addition to the clear endogenous regulation of BAP1 during trophoblast differentiation that demonstrate a significant downregulation of BAP1 as trophoblast acquires invasive properties. We have shown this by RT-qPCR on primary tissue samples, on human trophoblast cell lines that exhibit different invasive properties, on tissue sections by IHC, and on hTSCs upon differentiation into extravillous cytotrophoblast (EVT). These data are summarized in Figure 6 and Figure 6—figure supplement. Specifically, the newly added data are in Figure 6F, 6G and Figure 6—figure supplement 1B, and in the text on p. 12/13 lines 353-364.

Overall, these data clearly indicate that BAP1 gene function is conserved between mouse and human. The main focus of this manuscript is on describing a novel, key role of BAP1 in regulating EMT in trophoblast cells. While we conduct the majority of this study in the mouse, the above data strongly suggest that this function is conserved in human trophoblast. We feel that it is beyond the scope of the current study to unravel all aspects of the intricate BAP1/ASXL complex regulation in human placentation, which – in our opinion – would represent a separate manuscript. It should also be noted that these hTSC manipulation experiments are extremely challenging and state-of-the-art in the field; we attempted CRISPR KO and siRNA approaches, but these were not successful for technical reasons and the fragility of these cells. These approaches have not been established as routine experimental protocols in hTSCs and require substantial amounts of technological optimization, which is beyond the scope of this work and, again, would constitute a study in its own right. To the best of our knowledge, our overexpression approach represents the first-of-its-kind genetic manipulation of the recently derived hTSCs to confirm that a specific molecular mechanism is conserved between mice and humans.

2. The main takeaway from Figure 2 is that BAP1 is dispensable for mouse TSC maintenance and that BAP1 knockout results in increased expression of stem cell markers Cdx2 and Esrrb. Both of these findings were previously reported in the authors' 2018 paper (see Figure 4b in the Nature paper). Therefore, the statement that "BAP1 deletion does not impair the stem cell gene regulatory network" is not surprising and the authors should state clearly that these experiments confirm their prior observations.

We agree with the reviewers that we previously observed the upregulation of the stem cell markers *Cdx2* and *Esrrb* mRNA levels in *Bap1* KO mTSCs grown in stem cell conditions. We now go further by describing in detail the dynamics of the CDX2 and ESRRB protein and mRNA levels in stem cell conditions and during early differentiation in the absence of BAP1. Importantly, we find that deletion of BAP1 affects cell proliferation, and we also describe that BAP1 is primarily regulated by FGF signalling. We also investigate a number of additional marker genes to underscore the point of sustained stem cell potential. These additional data provide insights well beyond those previously reported.

In line with reviewers’ suggestion, we now state that in lines 185-191: “Previously, we reported that Bap1 deletion in mTSCs resulted in a upregulation of Cdx2 and Esrrb mRNA levels (Perez-Garcia et al., 2018). We assessed CDX2 and ESRRB protein levels in Bap1^-/-^ mTSCs compared to (empty vector) control cells across 24 hours of differentiation (0h=stem cell conditions; 4h, 8h, 24h = hours upon differentiation). The absence of BAP1 resulted in increased ESRRB and CDX2 protein levels confirming our previous observations”.

3. The overexpression data in Figure 4 is difficult to interpret. Vector transduced TSCs show a tight, epithelial morphology (Figure 3A), whereas the NT-sgRNA control cells look like they are undergoing EMT (Figure 4C). Why does the introduction of the NT-sgRNA induce EMT characteristics? Bap1 sgRNA1 cells seem less epithelial than the Vector transduced cells. Do NT-sgRNA TSCs have less BAP1 than Vector transduced TSCs?

We apologize that the photo of NT-sgRNA transfected cells was misleading. We have replaced the images with new examples that better represent the epithelial character of these cell lines (Figure 4B). Importantly, following the reviewers’ concern, we have compared the transcript levels of BAP1 in 3 different Vector control clones and the 3 replicates of NT-sgRNA mTSCs and find that they behave like close replicates, indistinguishable from each other (Panel A Author response image 1). In addition, we have also immunoblotted BAP1 (Panel B Author response image 1). The levels of BAP1 in Vector control compared to NT-sgRNA TSC are identical.

4. Moreover, all the data in Figure 4 are based on a single sgRNA that could activate BAP1 expression. To exclude off target effects, the authors should confirm the effect of BAP1 overexpression using another sgRNA or cDNA overexpression system.

We understand the reviewer’s concern about the off-target effects. However, the high specificity of the dCas9-SAM system has been proven by several studies to show extremely rare off-target effects (Thakore et al., 2015; Konermann et al., 2015; Saayman et al., 2016). In particular for the dCas9 technology, the frequency of off-target binding to functional exons is very low (Boyle et al., 2017), and any residual potential off-target effects are expected to be mitigated due to epigenetic protection (Hay et al., 2017; Cao et al., 2016).

The three sgRNAs used in this study were previously designed to minimize potential off-target effects (Joung et al., 2017, 2019) and target the 180bp region upstream of the target gene TSS, which represents the strongest predictor of activation efficiency (Konerman et al., 2015). In our case, the most effective guide was sgRNA1, the closest to the TSS. The other two sgRNAs did not significantly upregulate *Bap1* transcription. Only when they were transduced together was *Bap1* expression upregulated, presumably due to the contribution of sgRNA1 (Figure 4—figure supplement 1A).

Despite this well-reported evidence, following the reviewers’ suggestion, we have performed new experiments overexpressing BAP1 by viral transduction (see new Figure 4—figure supplement 1G, 1H) and show that the overexpression of BAP1 affects the regulation of genes involved in epithelial characteristics of mTSCs. These new data are entirely in line with the results obtained by CRISPR/Cas9-SAM overexpression and validate the immense efforts we have put into implementing this system in mTSCs, as shown in Figure 4. These corroborating results are described on p. 10 lines 279-283.

5. The authors need to examine the gene expression data more closely as well as the functional consequences of BAP1 overexpression on TSC proliferation and differentiation. In particular it would be important to compare the list of DEG in BAP1 KO and overexpression condition. Are they mirror-image or are there differences? For example, Zeb2 expression is strongly upregulated in BAP1 mutant line but not significantly altered in cells overexpressing BAP1. This should be discussed.

We would like to thank the reviewers for these suggestions as they have proven particularly helpful to better understand the specific gene network regulated by BAP1. We have now assessed the proliferation rate and performed time course differentiation experiments on *Bap1* CRISPR/Cas9-SAM overexpressing cells compared to control cells (new Figure 4C, 4G and Figure 4—figure supplement 1F) and find that the increased amount of BAP1 resulted in a slower proliferation rate and delayed differentiation towards TGCs. These data exactly mirror-image the higher proliferation rates and slower differentiation of *Bap1* KO TSCs (Figure 2 and Figure 2—figure supplement).

In addition, following the reviewers’ suggestions, we have compared the DEG in *Bap1* KO and overexpression conditions (Figure 4—figure supplement 1E and Supplementary Table 5) to narrow down the specific gene network regulated by BAP1. By looking at these opposing scenarios, we show that despite expected differences due to the different gene engineering system, the mirror-image behaviour of de-regulated genes clearly indicates that BAP1 levels critically regulate the epithelial state of trophoblast.

The new results are included in lines 253-254: “The upregulation of Bap1 resulted in tight epithelial mTSC colonies that proliferated at a slower rate than NT-sgRNA control mTSCs (Figure 4B and 4C)”.

Lines 266-272: “Intriguingly, there was substantial overlap between genes downregulated in Bap1 KO mTSCs and those upregulated in Bap1-overexpressing cells, and conversely also between genes upregulated in the KO and downregulated in the overexpressing cells. Thus, the two opposing models of Bap1 modulation (KO vs overexpression) provided mirror-image results that pivoted around the biological processes of epithelial cell integrity, cell adhesion and cytoskeletal organization (Figure 4—figure supplement 1E and Supplementary Table 5).”

Lines 276-285 “In line with the re-acquisition of epithelial properties, Bap1-overexpressing mTSCs exhibited a delay in differentiation towards the invasive TGC lineage and lower invasive capacity through Matrigel compared to NT-sgRNA control cells (Figure 4G, 4H and Figure 4—figure supplement 1F). Finally, we corroborated the data obtained by CRISPR/Cas9-SAM overexpression by performing exogenous GFP-Bap1 overexpression experiments in mTSCs grown in stem cell conditions, which similarly resulted in a significant upregulation of Cdh1 and strong downregulation of Cdh2, Zeb1, Zeb2, Snai1 and Vim expression (Figure 4—figure supplement 1G and 1H). These results demonstrate that precise levels of BAP1 regulate mTSC morphology, and that modulation of BAP1 levels affects the extent and speed at which trophoblast cells undergo EMT.”

6. In the abstract, the authors state that BAP1 function during trophoblast development is dependent on its binding to Asxl1/2/3. However, the data presented in this manuscript do not address whether BAP1 and Asxl1/2/3 are indeed part of the same complex in TSCs. Furthermore, the fact that Asxl1/2 KO increases expression of syncytial genes (Figure 5) does not provide direct evidence of functional synergy between these proteins and BAP1. This conclusion could be strengthened by demonstrating that Asxl1 and BAP1 indeed have a protein-protein interaction in TSCs and/or by deleting the BAP1 binding domain in Asxl1/2. It would also be instructive to examine whether the phenotype of BAP1 overexpression in TSCs (e.g. gain of epithelial features and reduced invasiveness) is dependent on Asxl1. This could be examined by overexpressing BAP1 in Asxl1-deficient TSCs.

We would like to thank the reviewers for this comment. Firstly, just to clarify, the deletion of *Asxl1* or *Asxl2* impaired the upregulation of syncytial genes (Figure 5F, 5G and Figure 5-supplement 1F). Beyond that, the fact that the *Asxl1* and *Asxl2* KO mTSC phenotype recapitulated to some extent the phenotype of *Bap1* KO mTSCs indeed suggested a functional synergy between BAP1 and ASXL proteins.

Following the reviewers’ suggestions, we have conducted additional experiments to assess the endogenous interaction of BAP1-ASXL proteins by co-immunoprecipitating reciprocally BAP1-ASXL1 and BAP1-ASXL2 in stem cell conditions (new Figure 5C and Figure 5—figure supplement 1C). In addition, to demonstrate that BAP1 function is dependent on interaction with ASXL, we examined the protein levels of ASXL1/2 in the presence or absence of BAP1. The absence of BAP1 did not affect the levels of ASXL1 or ASXL2 (new Figure 5—figure supplement 1D). Then, we performed siRNA experiments to show that the knockdown of *Asxl1* or *Asxl2* reduced BAP1 protein levels, indicating that the stability of BAP1 depends on its interaction with ASXL proteins. Moreover, these experiments allowed us to demonstrate that BAP1:ASXL1 is the predominant complex in stem cell conditions (Figure 5D and 5E).

The new results included in lines 298-310: “These results prompted us to investigate the nature of the BAP1-ASXL interaction in the mTSC context. To study the endogenous association of BAP1 and ASXL, we immunoprecipitated BAP1, ASXL1 and ASXL2 from extracts of mTSCs grown in stem cell conditions and tested for reciprocal interactions by WB. While we were not able to detect an association of BAP1 and ASXL2 in stem cell conditions, co-immunoprecipitation of BAP1 and ASXL1 revealed that BAP1:ASXL1 is the predominant complex in mTSCs (Figure 5C and Figure 5—figure supplement 1C). Then, we further analysed whether this interaction regulates the stability of the BAP1 and ASXL proteins. Whereas the absence of BAP1 did not affect the stability of ASXL1 and ASXL2 (Figure 5—figure supplement 1D), small interference RNA (siRNA)-mediated knockdown of either ASXL1 or ASXL2 resulted in a decrease of BAP1 protein levels (Figure 5D). This was particularly significant in the case of ASXL1 knockdown, in line with ASXL1 being the major complexing partner of BAP1 in stem cell conditions (Figure 5D).”

7. In some cases, experiments are carried out to "confirm" and "corroborate" hypotheses rather than test them. For example, the similarity between the gene expression signature of Bap1 mutant murine TSCs is and Bap1 mutant melanocytes and mesothelial cells is shown and emphasized. One wonders how unique is this similarity? Is Bap1 expression modulation observed in other EMT processes during development or in cancer? This should be explored and discussed.

Following the suggestion of the reviewers, we have explored and discussed whether the modulation of BAP1 regulates EMT during development and cancer in lines 399-412: “To the best of our knowledge, a direct link between BAP1 modulation and EMT regulation in early development and cancer has not been reported. However, results from previous studies of cancers that are strongly associated with an EMT process during metastatic transformation such as uveal melanoma, clear-cell renal cell carcinoma, gastric adenocarcinoma, colorectal cancer, and non-small-cell lung cancer showed a significant decrease in tumour BAP1 expression and worse clinical outcomes (Kalirai et al., 2014; Yan et al., 2016; Tang et al., 2013; Fan et al., 2012). On the background of our results reported here, it is tempting to speculate that the characteristic EMT and metastatic behaviour of these malignancies is induced by deletion or low abundance of BAP1. However, loss of BAP1 has also been reported to promote mesenchymal-epithelial transition (MET) in kidney tumours cells suggesting that its precise mode of function depends on the cell- and tissue-specific context (Chen et al., 2019). Further molecular analyses will be required to unravel the intricate regulation of the EMT pathway in different cellular contexts and the role of BAP1 in these processes.”

[Editors' note: further revisions were suggested prior to acceptance, as described below.]

The authors have substantially revised their manuscript in response to the reviewers' comments. This work significantly advances our understanding of trophoblast differentiation and in particular of Bap1/Asx1/2 complexes in regulation of EMT and invasive properties during this process, and should be of interest to the broad eLife readership. However, some conclusions should be reconsidered and the abstract clarity could be improved.One of the key concerns to the original manuscript was that the conclusion " the molecular mechanism by which BAP1 regulates the epithelial characteristics of trophoblast is conserved between mice and human." was not supported by experimental data. In the revised manuscript new data are presented that over expression of BAP1 by viral transduction in a new hTSC line promotes epithelial characteristics. The authors argue that additional loss-of-function experiments for human BAP1 gene go beyond the scope of the manuscript. Whereas this reviewer agrees, the current data supports but does not "show that the molecular function of BAP1 is conserved in mouse and humans.", as is stated in the Abstract. This conclusion should be toned down in the abstract. In fact the language used at the end of Introduction more appropriately aligns with the current experimental results (lines 128-131) The functional characterization of BAP1 in the human placenta and human trophoblast stem cells (hTSCs) suggests that the role of BAP1 in regulating trophoblast differentiation and EMT progression is conserved in mice and humans. "

Following the reviewer’s suggestion, we have reworded this sentence in the Abstract (lines 48-50): “Finally, both endogenous expression patterns and BAP1 overexpression experiments in human trophoblast stem cells suggest that the molecular function of BAP1 in regulating trophoblast differentiation and EMT progression is conserved in mice and humans.”

The clarity of the Abstract could be improved. In the sentence "Moreover, we show that this function is dependent on the binding of BAP1 to additional sex comb-like (ASXL1/2) proteins to form the Polycomb repressive deubiquitinase (PR-DUB) complex." it is not clear what "function" do the authors mean, as the previous sentence describes the effects of BAP1 protein downregulation from which BAP1 function (limiting EMT) needs to be deduced.

We have reworded this sentence in the Abstract for clarification (lines 45-47): “Moreover, we show that the function of BAP1 in suppressing EMT progression is dependent on the binding of BAP1 to additional sex comb-like (ASXL1/2) proteins to form the Polycomb repressive deubiquitinase (PR-DUB) complex.”

Line 367 "More significantly, however, the EMT markers CDH1, CLDN2" – CDH1 is not a key EMT marker, just the opposite. Please revise.

We have corrected the sentence in line 367 (now, in line 365): More significantly, however, the epithelial markers CDH1, CLDN2, TJP1 and VCL were significantly upregulated with a concomitant strong repression of mesenchymal marker genes CDH2 and ZEB2 (Figure 6F, 6G and Figure 6—figure supplement 1B).